# Confinement width and inflow-to-sediment discharge ratio control the morphology and braiding intensity of submarine braided channels: Insights from physical experiments and reduced-complexity models

Sam Y. J. Huang[1], Steven Y. J. Lai[1], Ajay B. Limaye[2], Brady Z. Foreman[3], Chris Paola[4]

[1]Department of Hydraulic and Ocean Engineering, National Cheng Kung University, Tainan, Taiwan
[2]Department of Environmental Sciences, University of Virginia, Charlottesville, Virginia 22904, USA
[3]Department of Geology, Western Washington University, Bellingham, WA 98225, USA
[4]Department of Earth Sciences, University of Minnesota, Minneapolis, Minnesota 55455, USA

*Correspondence to*: Steven Y. J. Lai (stevenyjlai@mail.ncku.edu.tw)

**Abstract.** Submarine channels conveying sediment gravity flows are often topographically confined, but the effect of confinement width on channel morphodynamics is incompletely understood. We use physical experiments and a reduced-complexity model to investigate the effects of confinement width ($B$) and inflow-to-sediment discharge ratio ($Q_{in}/Q_s$) on the evolution of submarine braided channels. The results show that a larger confinement width results in increased active braiding intensity ($BI_A$), and that $BI_A$ takes longer to stabilize (i.e., a longer critical time, $t_c$). At a fixed confinement width, a higher $Q_{in}/Q_s$ slightly decreases the $BI_A$. Digital Elevation Models of Difference (DoD) of the experiments allow measurement of the morphological active width ($W_a$) of the submarine channels and the bulk morphological change ($V_{bulk}$) within an experiment, defined as the sum of total erosion and deposition. We find that $W_a$ and $V_{bulk}$ are proportional to $B$. We further confirm that $BI_A$ is proportional to both dimensionless sediment-stream power ($\omega^{**}$) and dimensionless stream power ($\omega^*$). These trends are consistent for submarine braided channels both with and without confinement width effects. Furthermore, we built a reduced-complexity model (RCM) that can simulate flow bifurcation and confluence of submarine braided channels. The simulated flow distribution provides reliable predictions of flow depth and sediment transport rate in the experiments. Using kernel density estimation (KDE) to forecast the probability and trends of cross-sectional flow distribution and corresponding $BI_A$ under extreme events, we find that skewness of the flow distribution decreases as discharge increases. The development of braided submarine channels, shown here to extend to conditions of topographic confinement, suggests that factors not modelled here (e.g., fine sediment) may be necessary to explain the abundance of single-thread submarine channels in nature.

# 1 Introduction

High-resolution bathymetry and seismic data have revealed unprecedentedly detailed channels on submarine fans and deep-sea plains (Deptuck et al., 2007; Babonneau et al., 2010; Janocko et al. 2013). These submarine channels are mainly formed by sediment gravity flows, and the corresponding channel morphology and stratigraphy resemble their fluvial counterparts (Lajeunesse et al., 2010; Peakall and Sumner, 2015; Jobe et al., 2016). Most past studies of submarine channels have focused on meandering channels (Imran et al., 1999; Peakall et al., 2000; Keevil et al., 2006; Deptuck et al., 2007; Peakall et al., 2007; Straub et al., 2008; Sylvester et al., 2011). However, laboratory studies and field observations suggest that braided channels also develop in submarine environment and may develop morphologies similar to braided rivers (Belderson et al., 1984; Hesse et al., 2001; Foreman et al., 2015; Lai et al., 2017; Limaye et al., 2018).

For instance, experiments of laterally unconfined submarine braided channels were first confirmed at two laboratory facilities (Foreman et al., 2015), and demonstrated that density currents readily produce submarine braided channels for flow aspect ratios (depth-to-width) similar to those of fluvial braided rivers. The authors confirmed that the stability theory for river planform morphology (Parker, 1976) successfully describes submarine braided channels in both experiments and the field. Subsequent experiments (Lai et al., 2017) further demonstrated that laterally unconfined submarine braided channels (Fig. 1a) have similar responses to fluvial braided rivers (Egozi and Ashmore, 2009), for which both the active braiding intensity ($BI_A$) and total braiding intensity ($BI_T$) increase with increasing inflow discharge and bed slope. $BI_T$ is defined as the total number of channels in cross-section, while $BI_A$ is a subset of cross-sectional channel numbers, which reflects channels that can transport bed load, influence channel morphology and relate to channel bifurcations and avulsions (Ashmore, 2009; Egozi and Ashmore, 2009). Past fluvial and submarine braiding experiments (Ashmore 1991; Egozi and Ashmore, 2009; Bertoldi et al. 2009; Lai et al., 2017) show that $BI_A$ scales linearly with dimensionless stream power ($\omega^*$). While the recently proposed entropic braiding index (Tejedor et al., 2022) accounts for variation in the width of individual channel threads, in this study we use the traditional braiding index for ease of comparison to these previous studies.

In natural rivers, lateral confinement reduces braiding intensity (Garcia Lugo et al., 2015; Carbonari et al., 2020). In submarine settings, canyon walls, erodible terraces, and basement ridges could also limit the development of submarine braided channels. For instance, Orinoco deep-sea fan is the first reported field evidence of submarine braided channels and have a width-to-depth ratio around 60 to 70 (Belderson et al., 1984; Ercilla et al., 1998). Further downstream of these submarine braided channels, confinement by the Barbados deformation front and a basement ridge force the channels to transform into a single channel (Belderson et al., 1984; Callec et al., 2010). Along the Sicilian margin, submarine braided channels occur on the Stromboli slope valley (south eastern Tyrrhenian Sea) that is confined by a terrace and valley wall (Gamberi et al., 2011). In the same region, the Baia di Levante Fan (Vulcano Island, Italy), a modern submarine volcaniclastic fan, was formed by repetitive

stacking of gravity flows (Romagnoli et al., 2012), with clear erosive banks confining the observed submarine braided channels (Fig. 1b).

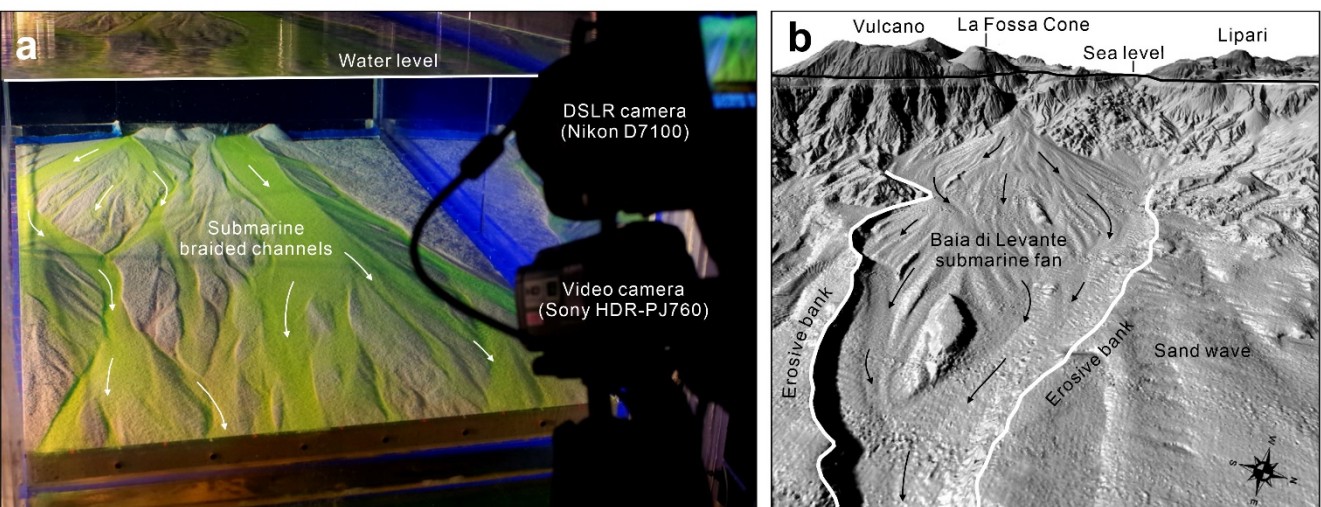

**Figure 1. Submarine braided channels: (a) in a laboratory-scale sedimentary basin, without lateral confinement (Lai et al., 2017); (b) on Baia di Levante submarine fan (Vulcano Island, Italy), with erosive banks as lateral confinement (modified from Romagnoli et al., 2012).**

No studies have yet investigated the evolution of submarine braided channels restricted by lateral confinement. In this study we investigate how submarine braided channels respond to confinement widths ($B$) and inflow-to-sediment discharge ratio ($Q_{in}/Q_s$) by using physical experiments and a reduced-complexity model. Our aims are: (1) to obtain high-resolution digital elevation models (DEMs) for the evolving topography of submarine braided channels; (2) to analyse the effect of confinement widths on active braiding intensity; (3) to develop a reduced-complexity model for predicting discharge distribution patterns; and (4) to establish scaling relationships across fluvial and submarine braided channel systems.

## 2 Methods

### 2.1 Experiments

The experimental basin (1.8 m long, 0.55 m wide, and 0.05 m deep) was submerged into a water tank (2.3 m long, 1 m wide, and 0.65 m deep) to investigate the effect of different confinement widths on the evolution of submarine braided channels (Fig. 2). The water tank provided still, ambient water to simulate a deep-sea environment. Three drains were set at the downstream end of the tank to remove turbid water. The initial bed slope ($S_0$) of the basin is adjustable. Prior to experiments, plastic sand (specific gravity = 1.5, median grain size $d_{50} = 0.34$ mm, uniformity coefficient $Cu = d_{60}/d_{10} = 1.64$) was formed into a planar bed within the basin. Then a straight channel was excavated in the middle with a fixed depth of 0.015 m and a prescribed

confinement width. At the upstream end, the same dry plastic sand was provided steadily by a motor-controlled conveyor belt to simulate bed load sediment input. Dyed and saturated saline inflows ($\rho_{in}$ = 1200 kg/m³) were supplied upstream to simulate long-lived, unconfined hyperpycnal flows or mud-rich turbidity currents (Métivier et al., 2005; Spinewine et al., 2009; Sequeiros et al., 2010; Weill et al., 2014; Foreman et al., 2015; Lai et al., 2016; 2017). Check-dam partitions at the inflow exits allowed the saline discharges and dry plastic sands to be fully mixed before entering to the basin.

Six runs of experiments were conducted to investigate the effects of different confinement width ($B$) and inflow-to-sediment discharge ratio ($Q_{in}/Q_s$) on the evolution of the submarine braided channels. Combinations of confinement $B$ (at 0.12 m, 0.24 m, and 0.48 m) and $Q_{in}/Q_s$ (at ~ 60 and ~90) encompass Series A and Series B (Table 1). The confinement widths were designed to distinguish from previous experiments (Foreman et al., 2015 and Lai et al., 2017) with full valley width ($W$), and the $B$ in this study is based on the proportion of 21.8%, 43.6% and 87.3% of $W$. The inflow unit width discharge is calibrated by $q_{in} = Q_{in}/b$, where $b$ is inflow width (not confinement width $B$), because the submarine braided channels do not occupy the entire confinement width. The inflow-to-sediment discharge ratio ($Q_{in}/Q_s$) and inflow unit width discharge ($q_{in}$) were controlled at the same time. Based on previous reported successful runs of Foreman et al. (2015) and Lai et al. (2017), $Q_{in}/Q_s$ = 60 and 90 are the two reasonable values of this ratio. The initial bed slope was kept at 0.114 (6.5 degrees, the most effective bed slope for forming submarine braided channels reported by Lai et al., 2017). Each experimental run lasted 100 min and was divided into 10 successive stages (10 min for each stage). During the experiments, time-lapse images were obtained using a Nikon D7100 at intervals of 5 sec. At the end of each stage, the inflow and sediment supply were stopped, and topography temporarily stabilized. Without draining the ambient water, the terrain surface was scanned. After that, a high-resolution (1 mm x 1 mm) digital elevation model (DEM), orthorectified image, and gradient map of each stage were constructed through a topographic imaging system (Lai et al., 2016; Lai et al., 2017).

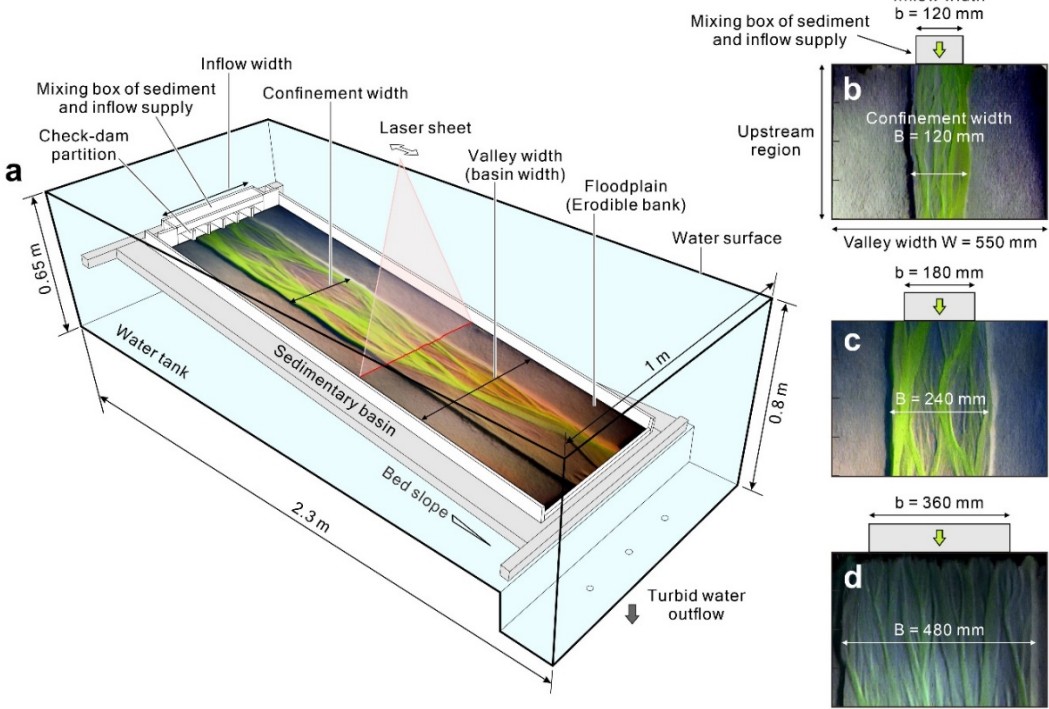

110

**Figure 2. (a)** Experimental setup for studying the evolution of submarine braided channels with different confinement widths and inflow-to-sediment discharge ratios.; **(b)** inflow width, confinement width and valley width of Runs A1 and B1; **(c)** inflow width, confinement width and valley width of Runs A2 and B2; **(d)** inflow width, confinement width and valley width of Runs A3 and B3.

115

**Table 1. Summary of experimental conditions.**

| Run | Inflow width $b$ (mm) | Confinement width $B$ (mm) | Valley width $W$ (mm) | Initial bed slope $S_0$ (°) | Inflow total discharge $Q_{in}$ (ml/s) | Sediment total discharge $Q_s$ (ml/s) | Inflow-to-sediment discharge ratio $Q_{in}/Q_s$ | Inflow unit width discharge $q_{in} = Q_{in}/b$ (mm²/s) | Final bed slope $S$ (-) | Critical time $t_c$ [s] | Averaged active braiding intensity at stable phase $BI_A$ |
|---|---|---|---|---|---|---|---|---|---|---|---|
| A1 | 120 | 120 | 550 | 6.5 | 6.61 | 0.109 | 61 | 55.11 | 0.1198 | 2400 | 1.5 |
| A2 | 180 | 240 | 550 | 6.5 | 10.02 | 0.171 | 59 | 55.68 | 0.1169 | 3600 | 2.5 |
| A3 | 360 | 480 | 550 | 6.5 | 19.95 | 0.314 | 64 | 55.42 | 0.1168 | 4800 | 4.3 |
| B1 | 120 | 120 | 550 | 6.5 | 10.13 | 0.113 | 89 | 84.42 | 0.1171 | 2400 | 1.1 |
| B2 | 180 | 240 | 550 | 6.5 | 15.27 | 0.167 | 91 | 84.86 | 0.1075 | 3600 | 2.1 |
| B3 | 360 | 480 | 550 | 6.5 | 30.08 | 0.332 | 91 | 83.56 | 0.1174 | 4800 | 3.6 |

## 2.2 Automated measurement of channel position

Braiding intensity has been widely used in fluvial braided rivers (Egozi and Ashmore, 2008; Bertoldi et al., 2009; Egozi and Ashmore, 2009) and applied to submarine braided channels (Lai et al., 2017; Limaye et al., 2018). In past analyses of submarine braided channels (Lai et al., 2017), the braiding intensity required manual interpretation of cross-sectional channel numbers. This approach is time-consuming and can be subjective. In this study, we developed an RGB color separation algorithm to identify channel positions, calculate cross-sectional channel numbers, and obtain an objective, repeatable, and averaged braiding intensity for a selected color image (Fig. 3). For instance, a total of 1080 color orthophotos were analyzed using an algorithm based on Eq. (1) that obtains enhanced gray-scale images:

$$\alpha G - \beta R - \gamma B \qquad (1)$$

where $\alpha$, $\beta$ and $\gamma$ are weighting coefficients to be calibrated by each run (e.g., $\alpha = 1$, $\beta = 0.8$, $\gamma = 0.35$ for Run A2). R, G, B represent the red, green and blue bands of a color image, respectively. The weighting coefficients used to generate the gray-scale images varied slightly between experimental runs due to different light settings and dye concentration. Then, the gray-scale images were converted into binarized images by an adaptive threshold method (Bradley and Roth, 2007) (Fig. 3b). After that the cross-sectional illumination values (from $x = 300$ to 1500 mm, extracted every 1 mm) were smoothed by a Gaussian filter to locate the channel positions (Fig. 3c to Fig. 3e, showing only three example cross sections). After the above procedure, the channel positions and cross-sectional channel numbers could be automatically obtained (Fig. 3f). Then the average value would be taken as the representative active braiding intensity ($BI_A$) of that image. The results of braiding intensity are presented in Section 3.2.

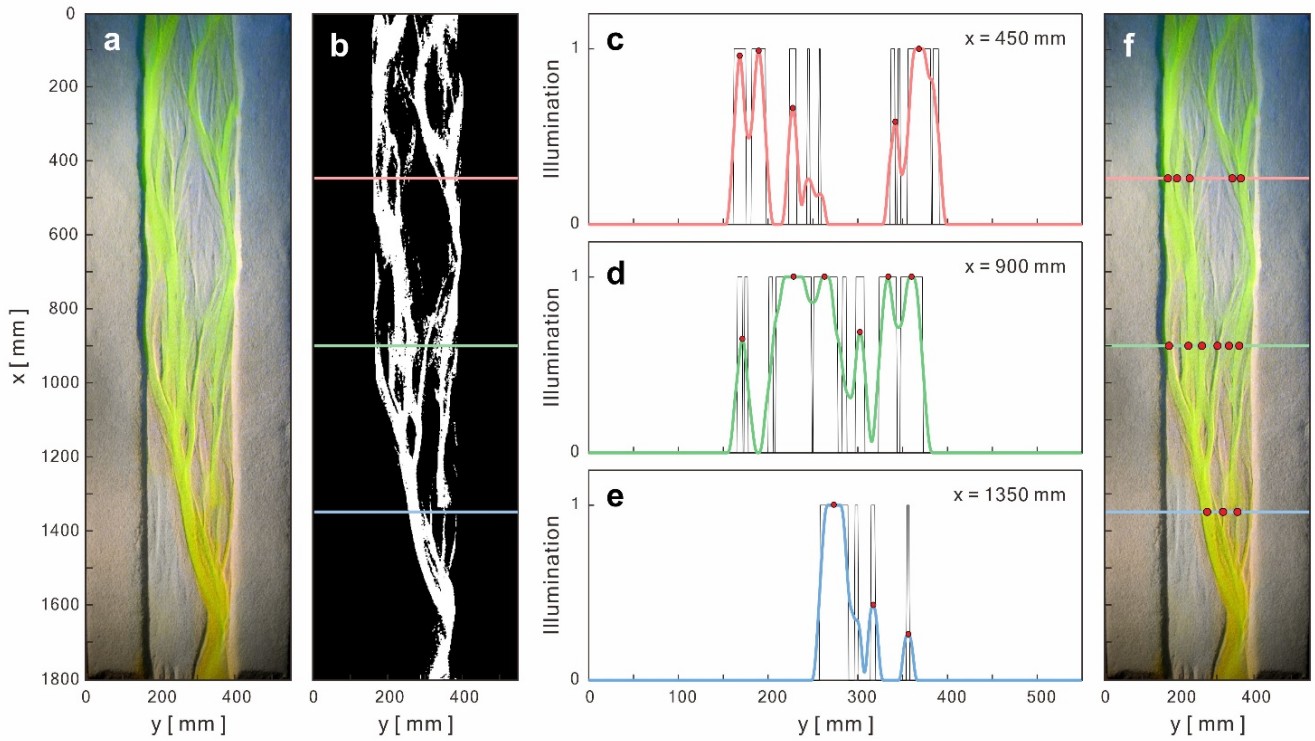

**Figure 3. Procedure of image processing for identifying channel positions for each experiment. (a) orthorectified color image; (b) converted binarized image; (c)-(e) extracted examples of cross-sectional illumination values and captured channel positions; (f) verification for the automatically captured channel positions and cross-sectional channel numbers.**

## 2.3 Reduced-complexity model

Reduced-complexity numerical models are often used to simulate macro-scale, long-term fluvial braided rivers (Murray and Paola, 1994; 1997; Thomas and Nicholas, 2002). The model proposed by Murray and Paola (1994, 1997) is the most influential work on reduced-complexity models. The model predicts the generic dynamics of stream braiding, including channel shifting, avulsion and migration. Later research modified their flow routing schemes to make reduced-complexity models comparable to shallow water equations models (Thomas and Nicholas, 2002; Thomas et al., 2007; Nicholas, 2009; Williams et al., 2016). Recently, the CAESAR-LISFLOOD model (Coulthard et al., 2013) was applied to map experimental submarine braided channels and bar geometry (Limaye et al., 2018). This analysis showed that compared to a subaerial braided channel with the same water and sediment fluxes, submarine braided channels developed fewer and deeper channels near the inlet, with declining topographic variations further downstream. In this study, we followed the numerical scheme of Thomas and Nicholas (2002) and developed a Matlab based reduced-complexity model, with modified hydraulic conditions for density currents. Our modifications assume large aspect ratios (width-to-depth ratio), reduced gravity and underflows entrain negligible ambient

water, and that the flows of submarine braided channels may resemble fluvial braided rivers (Foreman et al., 2015; Lai et al., 2017; Limaye et al., 2018).

The model starts calculations when the discharge is provided in the cell at the upstream boundary. Then the discharge is transferred downstream according to the calculation path of each row, and the model step ends when the calculation reaches the downstream boundary. The discharge is distributed from each upstream cell to the five immediate downstream cells (Thomas and Nicholas, 2002) rather than three cells used by Murray and Paola (1994, 1997). The discharge could be transferred laterally up to 60 degrees to the downstream direction. The allocated discharge in the five downstream cells is given by:

$$q_i = \frac{P_i}{\sum_{i=1}^{5} P_i} q_0 \tag{2}$$

where $q_0$ is the outflow discharge of the upstream cell; $P_i$ is the routing potential, which determines the proportion of discharge that can be allocated to the five downstream cells. In the five downstream cells, the $P_i$ of each cell was calculated from the upstream cell, starting from its water depth $h_0$. Assuming the density current is uniform, the flow depth at steady-state can be approximated by (Ippen and Harleman, 1952; Lofquist, 1960):

$$h_0 = C \left( \frac{\mu_{in} q_0}{Jg(\rho_{in} - \rho_a)S} \right)^{\frac{1}{3}} \tag{3}$$

where $C$ is a coefficient to be calibrated by each experiment ($C = 1.2$ for this study); $\mu_{in} = 1.68 \times 10^{-3}$ kgm$^{-1}$s$^{-1}$ is the dynamic viscosity of saline water at $20°$; $S$ is bed slope; $\rho_{in}$ is inflow density; $\rho_a$ is the density of ambient water; $J$ is a dimensionless parameter, which is proportional to the ratio of interface velocity ($V_i$) to maximum velocity ($V_{max}$) of the density current, given by:

$$\frac{V_i}{V_{max}} = \frac{12J-1}{12J^2+4J+\frac{1}{3}} \tag{4}$$

For a uniform density current at steady state, $V_i/V_{max} \cong 0.59$ (suggested by Ippen and Harleman, 1952). Therefore, the calculated $J$ value is 0.138 for this study. Flow depths ($h_i$) at each of the five downstream cells were approximated from:

$$h_i = h_0 + z_0 - z_i - S_0 dx \tag{5}$$

where $z_0$ is the bed elevation of the upstream cell; $z_i$ is the bed elevation of each of the five downstream cells; $dx$ is the distance from the upstream cell to each of the five downstream cells. If the $h_i$ calculated by Eq. (5) is a positive value, the flow can be routed; if $h_i$ is a negative value, the flow cannot be routed. Therefore, the values of $h_i$ affect the calculation method of $P_i$, which could be divided into the following two cases:

(1) When at least one of the 5 downstream cells have $h_i > 0$, then $P_i$ can be calculated. In the remaining cells, if its $h_i \leq 0$, its corresponding routing potential will be set to 0. $P_i$ is calculated as follows:

$$P_i = h_i^{1.67} S_i^{0.25} \tag{6}$$

where $S_i$ is the local bed slope between the cells that flow is being routed from and to; the slope exponent 0.25 was suggested by Thomas and Nicholas (2002).

(2) If all the five downstream cells have $h_i \leq 0$, the flow is assumed to be critical. This behavior often happens when density currents flood over sandbars, which is more likely to occur in submarine braided channels than in fluvial braided rivers (Sequeiros, 2012). Flow depths ($h_i$) at each of the five downstream cells are then approximated by:

$$h_i = h_{max} + z_{min} \tag{7}$$

where $h_{max}$ is the maximum depth, $h_{max} = \left(q_0/\sqrt{g'}\right)^{0.67}$; $g' = (\rho_{in} - \rho_a)g/\rho_{in}$ is reduced gravity; $z_{min}$ is the lowest bed

elevation in the downstream cell. The corresponding $P_i$ is:

$$P_i = h_i^{1.5} \tag{8}$$

The above rules together constitute our reduced-complexity model for simulating the flows of submarine braided channels. In our simulations, the initial bed condition is imported from the experimental DEM. The simulated inflow discharge is uniformly distributed at the upstream cells based on the experimental total inflow discharge. Notice that our reduced-complexity model

only simulates the underflows at the stable phase (Stage 6 to Stage 10), i.e., the morphological changes are excluded. Additionally, extra simulations are performed with double inflow discharge based on the experimental DEM at the final stage. These simulations are used for testing whether the linear relationship still holds between dimensionless stream power and active braiding intensity under extreme events. These results will be addressed in Section 3.4 and Section 4.

## 3 Results

**3.1 Morphological evolution of submarine braided channels**

Physical experiments document the initiation and evolution of submarine braided channels in response to set, known input parameters and conditions (Fig. 4). For instance, in Run A2, the initial confinement width was set at $B = 0.24$ m, and the inflow-to-sediment discharge ratio was $Q_{in}/Q_s = 58.7$. By $t = 1200$ s, two small channels and a few sandbars appeared within the straight channel (Fig. 4b). At $t = 2400$ s, multiple channels and sandbars gradually developed downstream. At $t = 3600$ s,

the flows showed frequent bifurcation and convergence. At $t = 4800$ s, slight bank erosion appeared on both sides and some channels were abandoned. At $t = 6000$ s, channels frequently change their routes and sandbars grew and elongated. The DEM hillshades clearly document the channel relief and bar thickness of submarine braided channels (Fig. 5). These high-quality DEMs provided the basis for subsequent morphometric analyses.

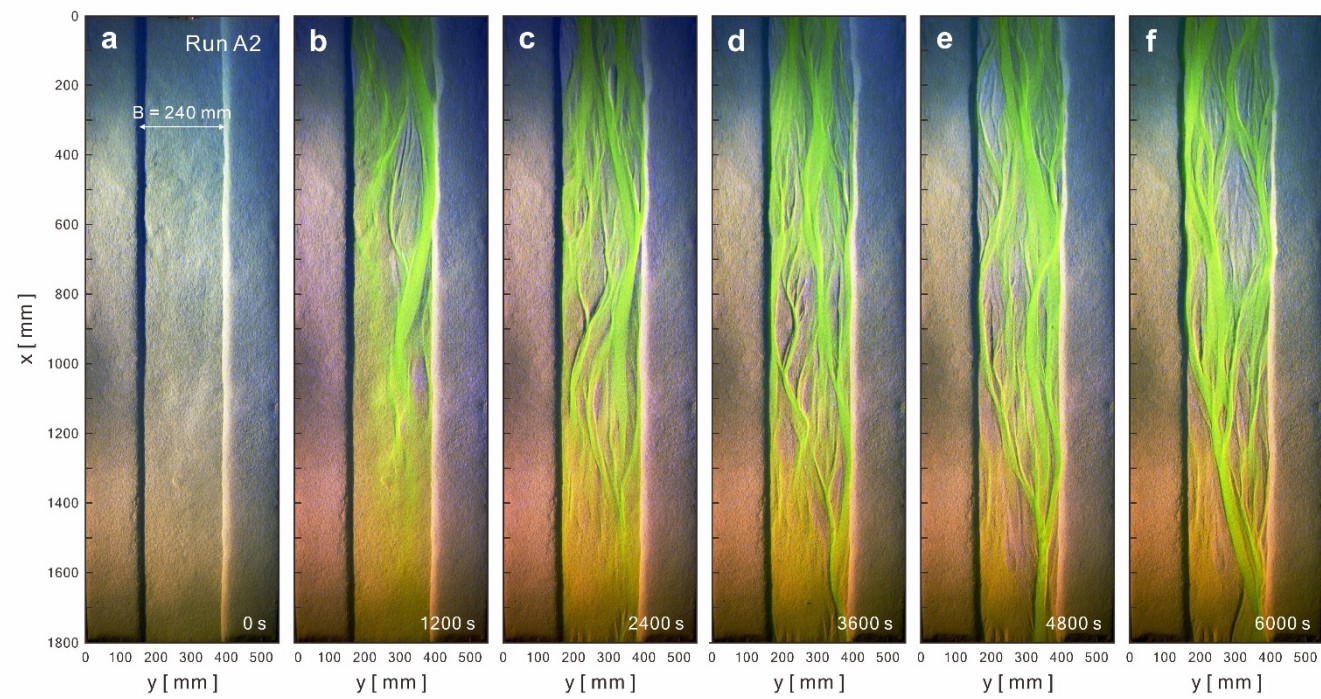


**Figure 4. Evolution of submarine braided channels of Run A2 from $t = 0$ to 6000 s. The confinement width is $B = 240$ mm. The inflow unit width discharge is $q_{in} = 55.68$ mm$^2$/s. The inflow-to-sediment discharge ratio is $Q_{in}/Q_s = 59$.**

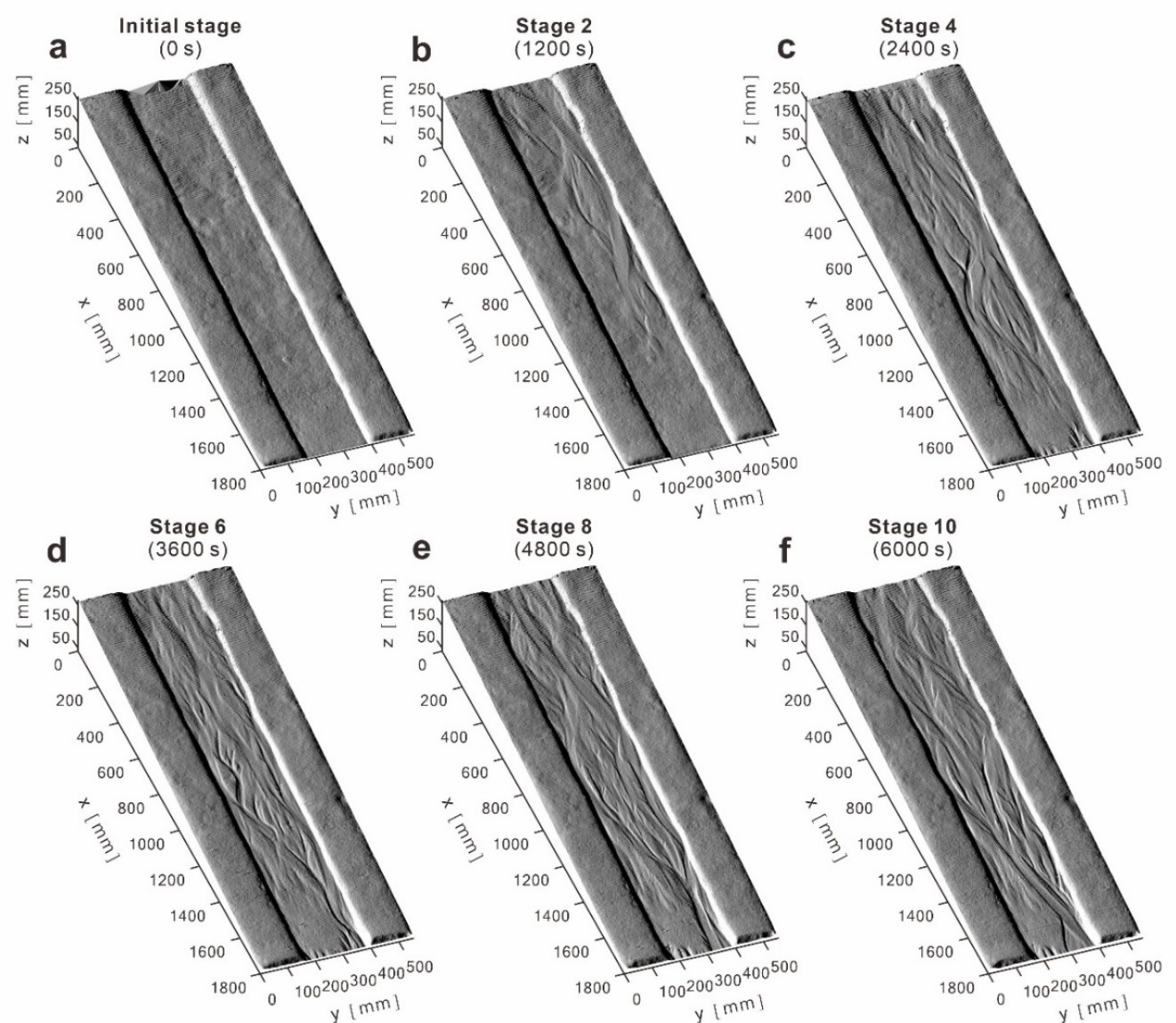

**Figure 5. DEM hillshades for Run A2 from $t = 0$ to 6000 s.**

Amongst Series A and Series B (Fig. 6), under the same $Q_{in}/Q_s$ condition, a higher confinement width produces more channels and higher degree of braiding. On the contrary, when confinement width is fixed, a larger $Q_{in}/Q_s$ generates fewer channels but wider main channels. For instance, in Series A ($Q_{in}/Q_s = 60$), Run A3 (Fig. 6c) has more channels and higher degree of

braiding. Similarly, in Series B ($Q_{in}/Q_s = 90$), Run B3 (Fig. 6f) is more braided. However, when confinement width is fixed (e.g., $B = 0.24$ m, Fig. 6b and Fig. 6e), a larger $Q_{in}/Q_s$ would have less channels but wider main channels. These characteristics are also valid for confinement $B = 0.12$ m (Fig. 6a and Fig. 6d) and $B = 0.48$ m (Fig. 6c and Fig. 6f). Additionally, a larger

$Q_{in}/Q_s$ also makes sand bars more connected and facilitates small sandbars to merge into longer and larger sandbars (Fig. S1 to Fig. S10 and Movie S1 to Movie S6 provide more detailed evolution processes).


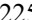

**Figure 6. The last orthophoto at $t = 6000$ s for Series A ($q_{in} \cong 55$ mm²/s, $Q_{in}/Q_s \cong 60$) and Series B ($q_{in} \cong 84$ mm²/s, $Q_{in}/Q_s \cong 90$).**

## 3.2 Evolution of active braiding intensity

The evolution of active braiding intensity reflects the critical time for submarine braided channels to reach a stable phase and the influence of confinement width and inflow-to-sediment discharge ratio (Fig. 7). For instance, the evolution of the braiding index over time for each run included two phases: (1) a rising phase and (2) a stable phase. We interpret the time for development of the stable phase as a critical time ($t_c$). Our results showed that $t_c$ is proportional to $B$, e.g., when $B = 0.12$ m, $t_c = 2400$ s; when $B = 0.24$ m, $t_c = 3600$ s; when $B = 0.48$ m, $t_c = 4800$ s. In addition, when $Q_{in}/Q_s$ is fixed, $BI_A$ increases as $B$ increases, e.g., Series A (Fig. 7a to Fig. 7c) and Series B (Fig. 7d to Fig. 7f). On the contrary, when $B$ is fixed, $BI_A$ at the stable phase decreases slightly as $Q_{in}/Q_s$ increases, e.g., $BI_A$ of Series B at $t = 6000$ s are all slightly smaller than that of Series A. Therefore, a larger confinement width postpones the critical time and increases the active braiding intensity at the stable phase. However, under the same confinement width, a larger inflow-to-sediment discharge ratio slightly decreases the $BI_A$.

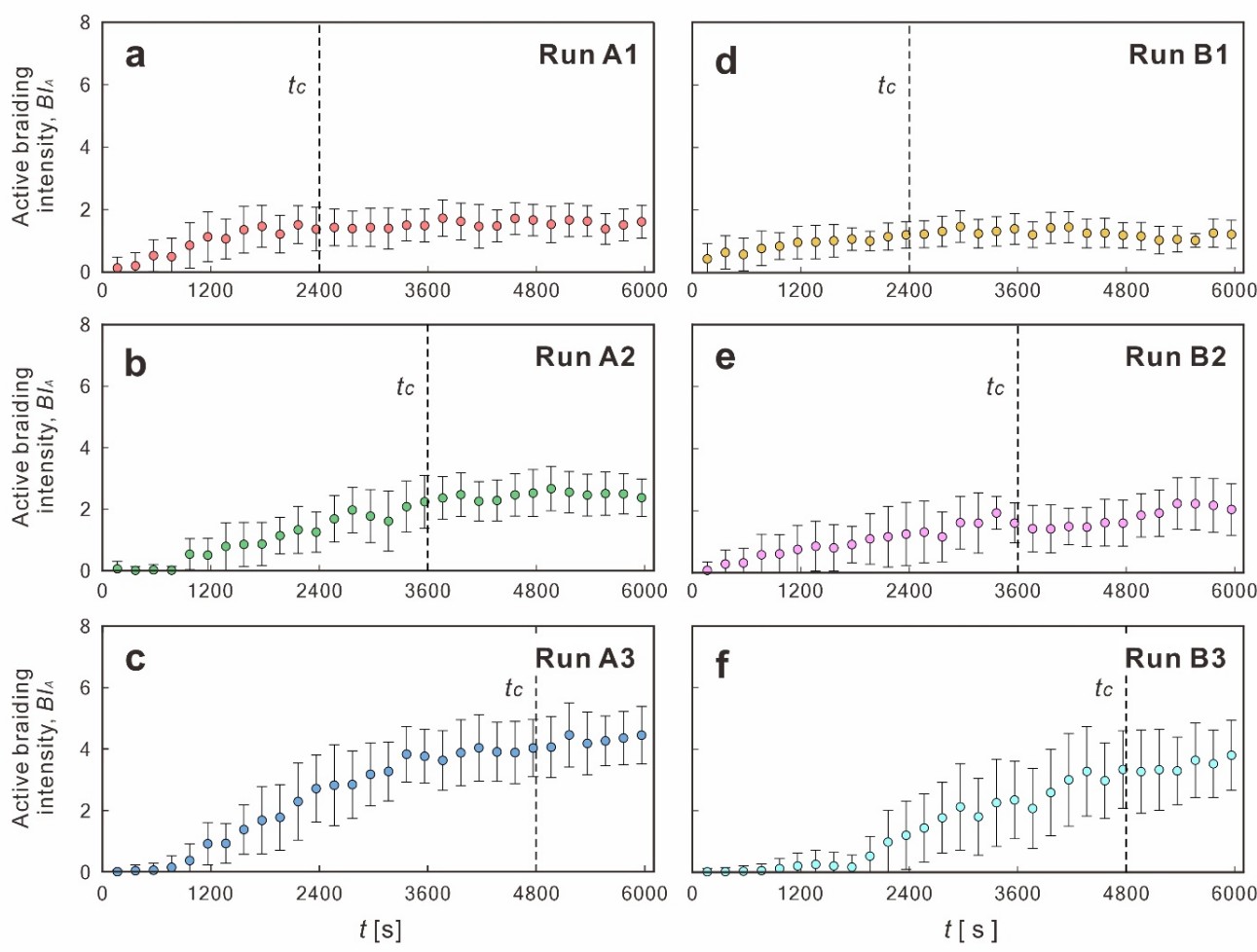


**Figure 7. Evolution of active braiding intensity ($BI_A$) for (a)-(c) Runs A1, A2 and A3; (d)-(f) Runs B1, B2 and B3; $t_c$ is the critical time for $BI_A$ reaching a stable phase.**

### 3.3 Volume change and morphological active width

DEM of Differences (DoDs) demonstrate the sediment distribution patterns of erosion and deposition for submarine braided channels (Fig. 8). The results showed that confinement width and bank erosion was not proportional for Series A or Series B. For instance, when $B = 0.24$ m, bank erosion on both sides was the most significant, especially for Run B2 (Fig. 8e). On the contrary, when confinement width is fixed, a larger $Q_{in}/Q_s$ would cause more severe bank erosion. Moreover, in Series B (Fig. 8d to 8f), the areas of erosion and deposition became more continuous and contiguous. In Series A ($q_{in} \cong 55$ mm$^2$/s, $Q_{in}/Q_s \cong$
60), most of the submarine braided channels develop within the given confinement width without much widening to accommodate the inflow. However, in Series B ($q_{in} \cong 84$ mm$^2$/s, $Q_{in}/Q_s \cong 90$), the submarine braided channels would widen

the given confinement width to accommodate the larger inflow and cause stronger bank erosion. Besides, the DoDs also reveal the morphological active widths ($W_a$) and bulk changes ($V_{bulk}$) for each run. The relationship between $W_a$ and $V_{bulk}$ for both fluvial and submarine braided channels will be discussed in Section 4.


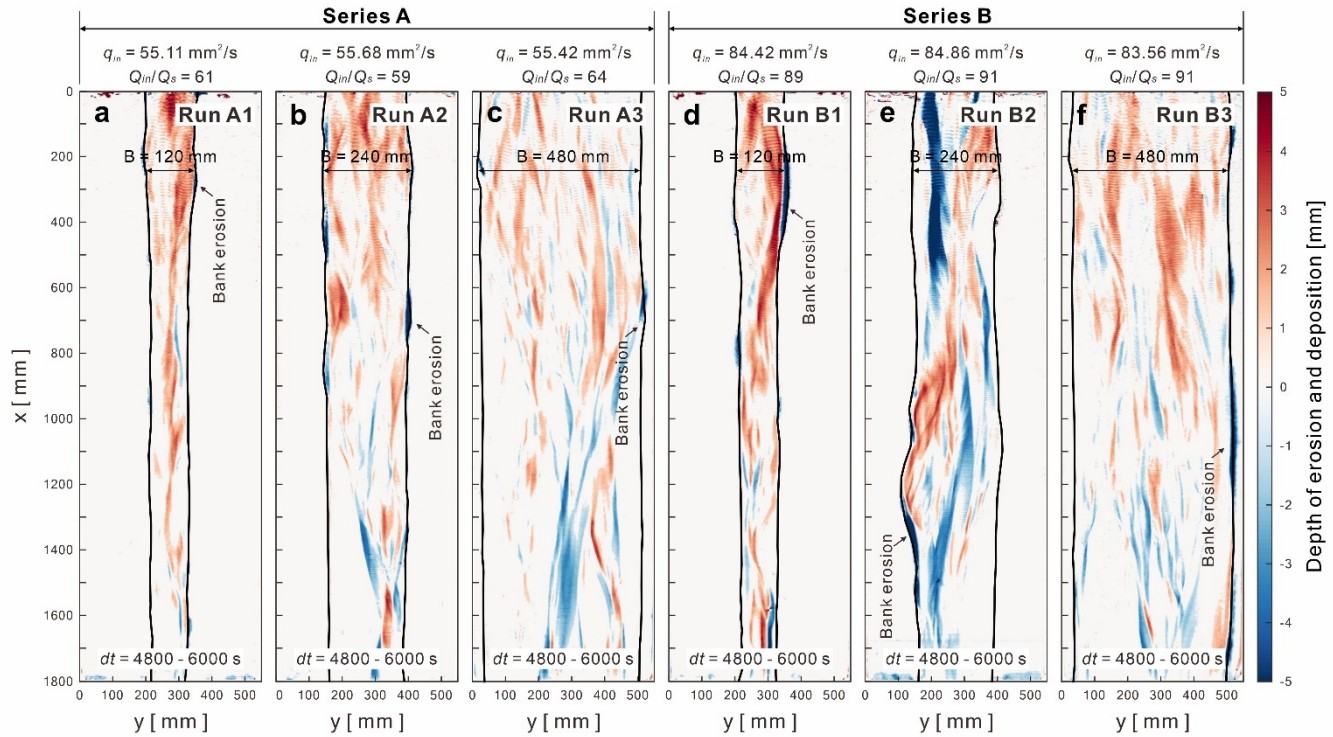

Figure 8. The DEM of difference (DoDs) for Series A ($q_{in} \cong 55$ mm²/s, $Q_{in}/Q_s \cong 60$) and Series B ($q_{in} \cong 84$ mm²/s, $Q_{in}/Q_s \cong 90$). The time duration for DoD calculation is $dt = 4800\text{-}6000$ s (last 20 min).


The normalized cross-sectional volume changes can further quantify the heterogeneity of the DoDs (Fig. 9). We extracted the DoD values on each cross-section (from $x = 0$ to 1800 mm, every 1 mm), calculated the sum of its volume (positive value represents deposition; negative value indicates erosion), and normalized it by the absolute value of total volume change (the sum of the entire DoD values). The results show that the medians calculated by all runs are greater than 0, indicating that the

DoD of each run is net depositional. When $B = 0.12$ m (Run A1 and Run B1), the DoDs are relatively homogeneous, i.e., the maximum and minimum values are closer to the median, showing a symmetrical distribution, but there are more outliers. When $B = 0.48$ m (Run A3 and Run B3), the DoDs are relatively heterogeneous, i.e., the maximum and minimum values are far from the median, showing an asymmetric distribution, but less outliers. These trends hold for both $Q_{in}/Q_s = 60$ (Series 1) and 90 (Series 2). Therefore, we propose that confinement width and heterogeneity of DoD are positively correlated.

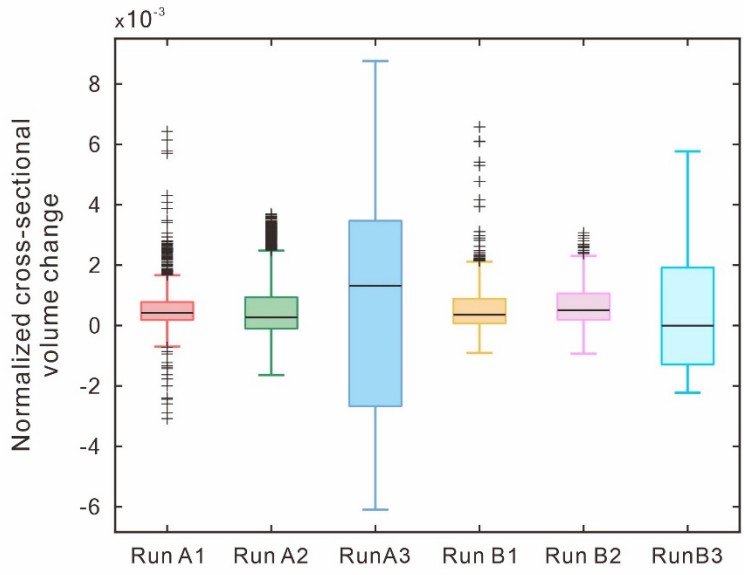


**Figure 9. Normalized cross-sectional volume changes for all runs.**

### 3.4 Simulated flow discharge and sediment transport rate

The reduced-complexity model for simulating submarine braided channels captured the pattern of flow bifurcation and

confluence (Fig. 10). Modeled flow occupied the main channels that developed in each run of the experiment. In Series A (Fig. 10a to Fig. 10c), a larger confinement width increased the number, width and discharge of the main channels. In Series B (Fig. 10d to Fig. 10f), the main channels became wider and more significant. The simulated flow characteristics are similar to those qualitatively observed in the experiments. However, validating the reduced-complexity model by directly comparing the measured flow depths is unfeasible because the averaged depth of experimental density currents is less than 3 mm. Therefore,

we converted the simulated and experimental flow maps (Fig. S11 to Fig. S16) into binarized images and compared their pixel differences. We examined the cross-sectional differences between the experimental and simulated flows every 1 mm along the $x$ direction. The results (Fig. S17) showed that the errors are less than 30 % for $B = 0.48$ m (Run A3 and Run B3) and even less than 6.5 % for $B = 0.12$ m (Run A1 and Run B1). This indicates that our reduced-complexity model is a reliable and applicable tool for simulating submarine braided channels.

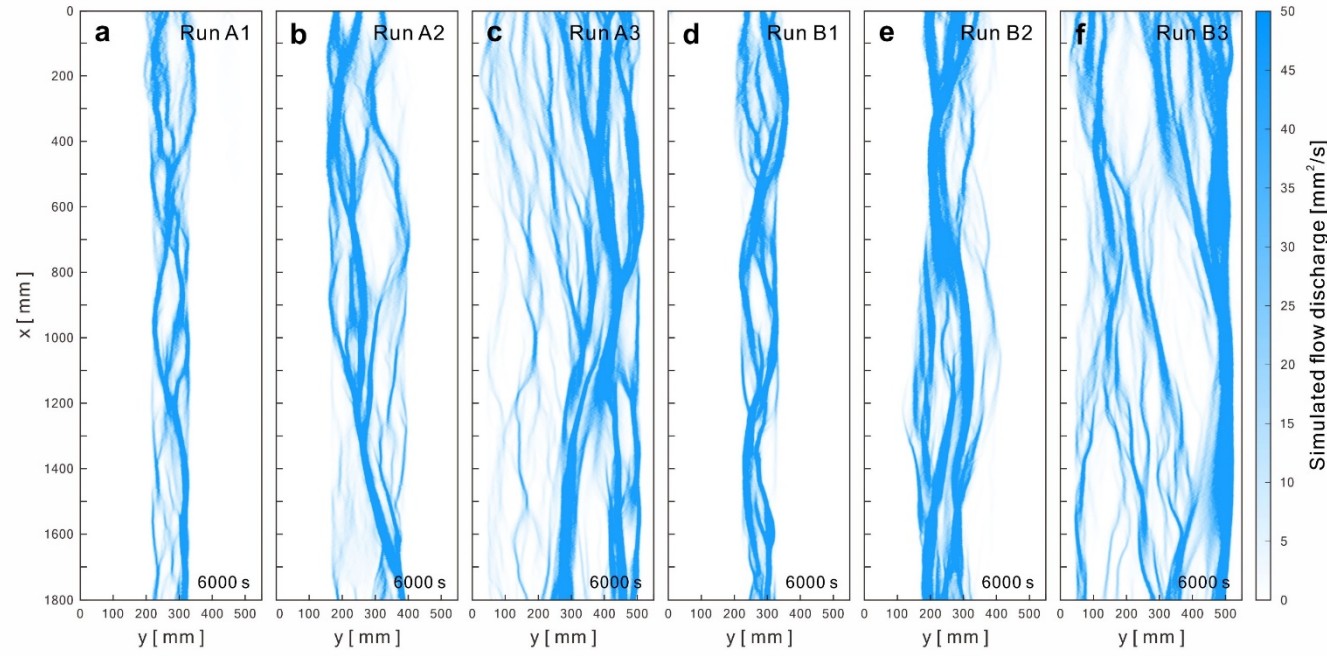

**Figure 10. Submarine braided channels simulated by the reduced-complexity model for each run at $t = 6000$ s.**

The simulated flows extracted at the midstream illustrates the discharge, flow depth and sediment flux distribution patterns affected by different confinement widths (Fig. 11). For instance, for Series A the modality of main channels belongs to unimodal distribution (Fig. 11a to Fig. 11c); for Series B the modality of main channels includes bimodal and multimodal distributions (Fig. 11d to Fig. 11f). Combining the simulated flow discharges and the bed load equation of Schoklitsch (1950), the sediment flux (kg/m-s) for each five downstream cells could be roughly estimated by:

$$q_{bi} = 2500 S_i^{3/2} (q_i - q_c) \tag{9}$$

where $S_i$ is local bed slope; $q_i$ is flow discharge (m²/s); $q_c$ is critical discharge (m²/s), given by:

$$q_c = 0.26(G - 1)^{5/3} \left( \frac{d_s^{3/2}}{S_i^{7/6}} \right) \tag{10}$$

where $G$ is the specific gravity of sediment ($G = 1.5$ for plastic sand); $d_s$ is medium grain size ($d_s = 0.34$ mm for this study), which is suitable for the grain size between 0.31 to 7.02 mm; Note that there is no sediment flux when $q_i < q_c$.

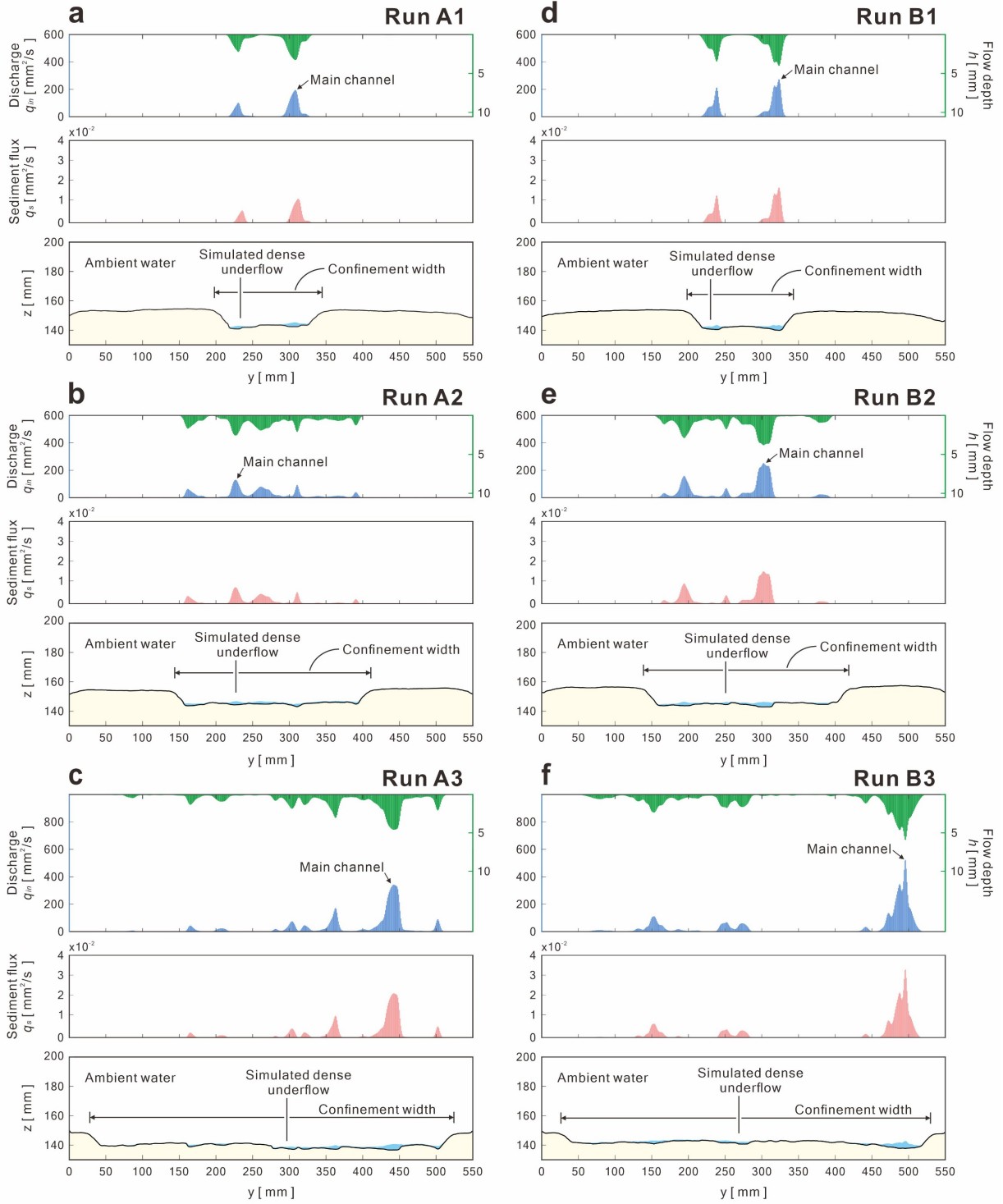

Figure 11. Simulated discharge, flow depth and sediment flux extracted at the midstream ($x$ = 900-1000 mm) for each run. The initial bed surface is based on the experimental DEM at the final stage.

Kernel density estimation (KDE) confirms the probability distribution of simulated flows are robust and consistent. Under extreme events, the KDE provides the predicted pattern of cross-sectional discharge and corresponding $BI_A$ (Fig. 12). For instance, we extracted the simulated flow discharge at the middle reach ($x$ = 950 to 1050 mm) to perform probability density function (PDF) and KDE statistical analyses. We excluded the data for which its unit discharge is less than 50 mm$^2$/s to avoid too many outliers. The KDE shows that the PDF of simulated discharges are all positively skewed and less sensitive to confinement width. On the contrary, when doubling the inflow discharge, the skewness of KDE decreases, allowing the given discharges be allocated over a broader unit discharge. Therefore, the number of channels and corresponding $BI_A$ all increased. Our reduced-complexity model appears consistent with the behavior of submarine braided channels, although the model still requires calibration and validation with more field bathymetric surveys.

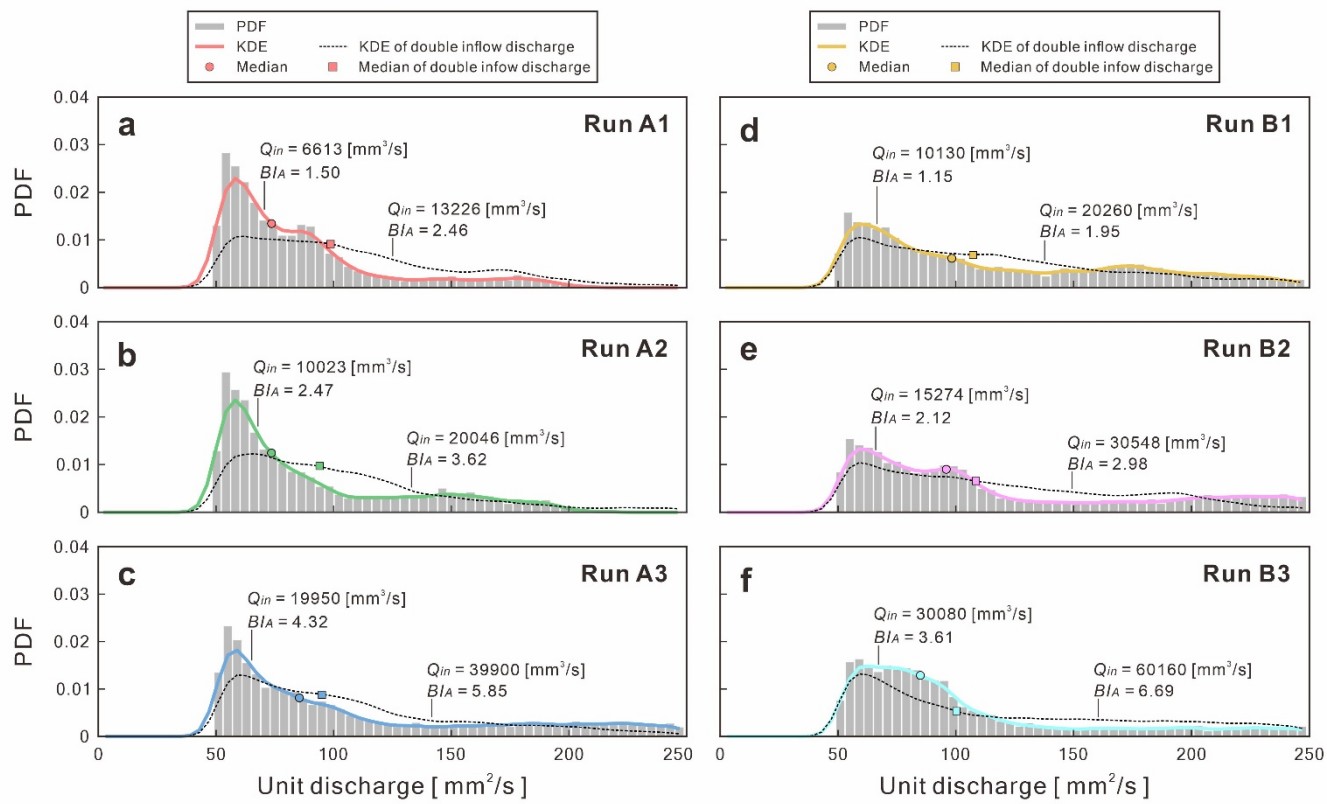

**Figure 12. Kernel density estimation (KDE) for the probability density function (PDF) of simulated flow distribution for each run. Color thick lines represent the KDEs of different runs. Black dash lines are the KDEs of double inflow discharge.**

## 4 Discussion

The purpose of this study is to understand the effects of confinement width and inflow-to-sediment discharge ratio on the
evolution of submarine braided channels using physical experiments and reduced-complexity models. We propose that confinement width has a stronger influence than inflow-to-sediment discharge ratio on the morphological evolution of submarine braided channels. The scaling relationships established by experiments and simulations can help us better interpret the evolution of field-scale submarine braided channels.

First, we find that both active width ($W_a$) and bulk change ($V_{bulk}$) are proportional to confinement width ($B$). Under a fixed reach length, a wider $B$ would generate a larger active area. Then, the larger active area would give a larger active width. By definition (Ashmore et al., 2011; Peirce et al., 2018), bulk change is the sum of the absolute values of erosion volume ($V_e$) and deposition volume ($V_d$) in a DoD. A wider $B$ increases the non-zero areas in a DoD, therefore increase the $V_{bulk}$. For example, the $W_a = 0.12-0.14$ m for $B = 120$ mm (Fig. 13); the $W_a = 0.24-0.28$ m for $B = 240$ mm; the $W_a = 0.48-0.49$ m for $B = 480$
mm. Our results show that $W_a$ is proportional to $V_{bulk}$, which agrees with the response of fluvial braided rivers (gray dots in Fig. 13). Even if the Reynolds number ($Re$) and densimetric Froude number ($Fr_d$) of our submarine braided channels indicate that flows in these channels are laminar and subcritical (Table 2), as long as the depth-to-width ratio ($h/B$) decreases and slope-to-densimetric Froude number ratio ($S/Fr_d$) increases, submarine braided channels will naturally form (Fig. 14). Our results agree with the morphological classification of fluvial rivers (Parker, 1976). In short, the active width and bulk change
of both submarine and fluvial braided channels share the same trend. This implies that the two braiding systems may have similar physical mechanisms.

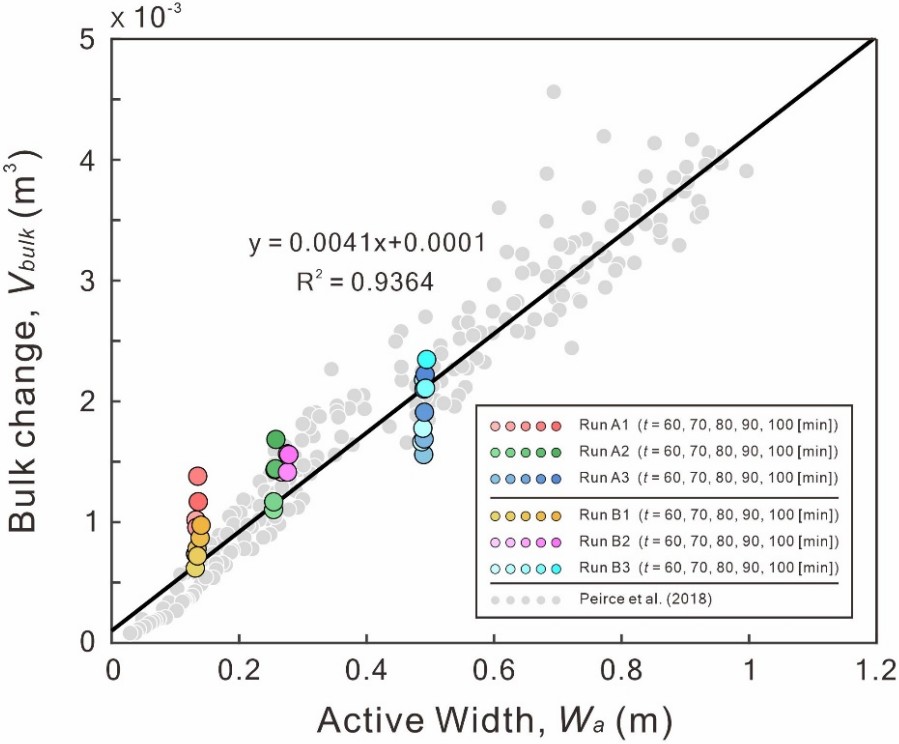

**Figure 13. The trend of active width ($W_a$) to bulk change ($V_{bulk}$). Color dots represent data of submarine braided channels (this study). Gray dots are experimental data of fluvial braided rivers (Peirce et al., 2018).**


**Table 2. Physical parameters and dimensionless numbers related to the submarine braided channels.**

| Run | Flow depth $h$ (mm) | Flow velocity $u$ (mm/s) | Reynolds number $Re$[1] | Densimetric Froude number $Fr_d$[2] | Dimensionless critical time $t_c^*$[3] | Reduced bed shear stress $\tau_b$[4] | Shields parameter $\theta$[5] | Boundary Reynolds number $Re_p$[6] | Dimensionless Stream Power $\omega^*$[7] | Dimensionless sediment-stream power $\omega^{**}$[8] |
|-----|------|------|------|------|------|------|------|------|------|------|
| A1 | 2 | 27.55 | 55.11 | 0.482 | 4.038E+08 | 0.470 | 0.470 | 16.48 | 106.8 | 0.720 |
| A2 | 2 | 20.88 | 41.76 | 0.365 | 9.180E+08 | 0.459 | 0.458 | 16.28 | 157.9 | 1.419 |
| A3 | 2 | 20.78 | 41.56 | 0.363 | 2.436E+09 | 0.458 | 0.458 | 16.28 | 313.5 | 2.509 |
| B1 | 2 | 42.21 | 84.42 | 0.738 | 6.186E+08 | 0.460 | 0.459 | 16.30 | 159.9 | 0.493 |
| B2 | 2 | 31.82 | 63.64 | 0.556 | 1.399E+09 | 0.422 | 0.422 | 15.61 | 221.3 | 0.958 |
| B3 | 2 | 31.33 | 62.67 | 0.548 | 3.674E+09 | 0.461 | 0.460 | 16.32 | 475.9 | 1.761 |

[1]Reynolds number, $Re = \frac{hu}{v} = \frac{Q}{v}$, where $h$ is the flow depth of the saline underflow, estimated from the experiments ($h \approx 2$ mm); $u = Q_{in}/(hB)$ is the cross-sectional averaged flow velocity; $Q_{in}$ is inflow total discharge; $B$ is confinement width; $v$ is the kinematic viscosity of water ($v = 10^{-6}$ m$^2$s$^{-1}$ at $20°$ C).

[2]Densimetric Froude number, $Fr_d = \frac{u}{\sqrt{g'h}}$, $g' = g(\rho_{in} - \rho_a)/\rho_{in}$ is the reduced gravity; $\rho_{in}$ is the density of inflow ($\rho_{in} = 1200$ kg/m$^3$); $\rho_a$ is the density of ambient water ($\rho_a = 1000$ kg/m$^3$).

[3]Dimensionless critical time, $t_c^* = t_c Q_{in}/d_s^3$, where $t_c$ is critical time; $d_s$ is sediment grain size ($d_s = 0.34$ mm).

[4]Reduced bed shear stress, $\tau_b = (\rho_{in} - \rho_a)ghS$, where $S$ is bed slope.

[5]Shields parameter, $\theta = \frac{\tau_b}{(\rho_s - \rho_{in})gd_s} = \frac{(\rho_{in} - \rho_a)ghS}{(\rho_s - \rho_{in})gd_s} = \frac{hS}{R'd_s}$, where $\rho_s$ is the density of plastic sand ($\rho_s = 1500$ kg/m$^3$); $R' = (\rho_s - \rho_{in})/(\rho_{in} - \rho_a)$.

[6]Boundary Reynolds number, $Re_p = \frac{u_* d_s}{v}$, $u_* = \sqrt{\tau_b/(\rho_{in} - \rho_a)}$ is shear velocity.

[7]Dimensionless stream power, $\omega^* = \frac{(\rho_{in} - \rho_a)QS}{\rho_{in} w_s d_s^2}$, $w_s = \frac{Rgd_s^2}{C_1 v + (0.75 C_2 Rgd_s^3)^{0.5}}$ is sediment settling velocity reported by *Ferguson and Church* [2004]; $R = (\rho_s/\rho_{in} - 1)$ is submerged relative density of sediment ($R = 0.25$ in this study); $C_1 = 18$ and $C_2 = 1$ are two constants for typical natural sands.

[8]Dimensionless sediment-stream power, $\omega^{**} = \left(\frac{B}{h}\right)\left(\frac{Q_s}{Q_{in}}\right)S^{0.2}$ (modified from Paola, 2001).

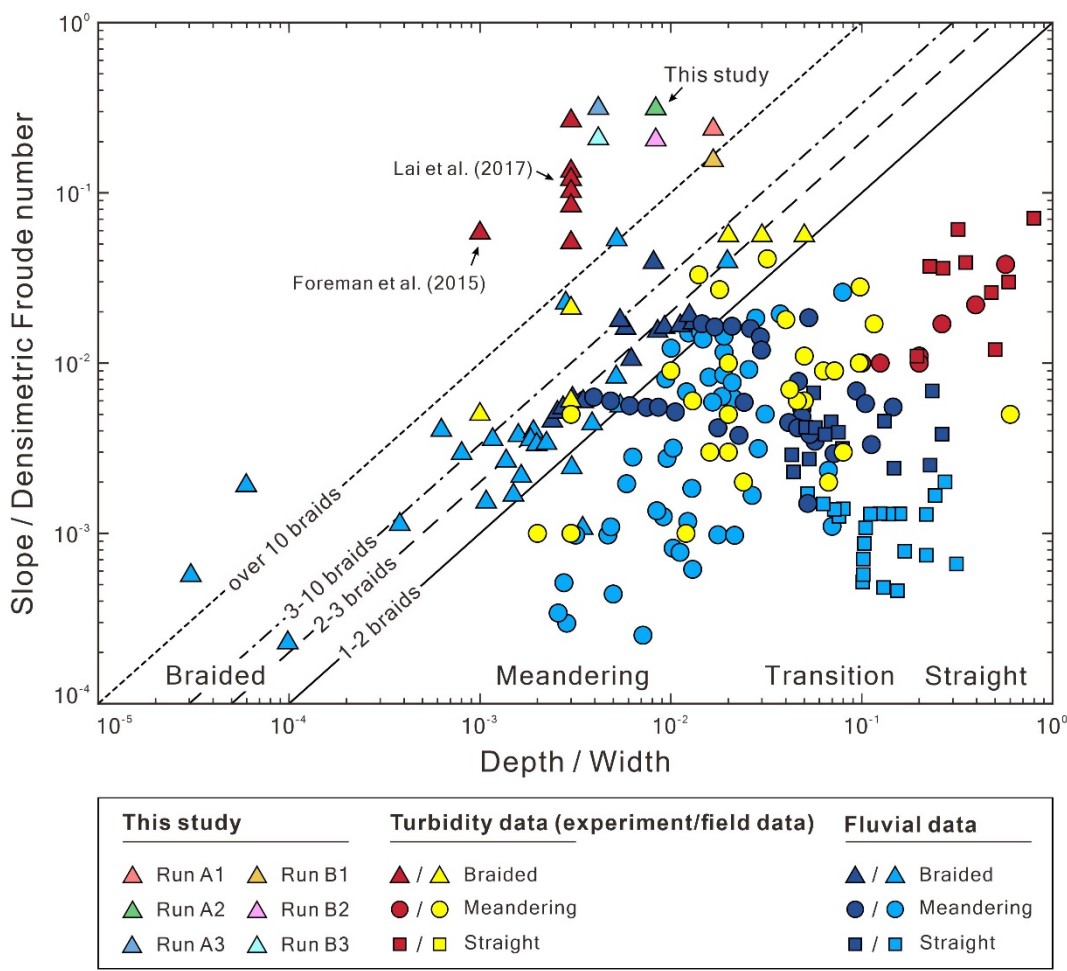


**Figure 14. Morphological classification of fluvial rivers and submarine channels (modified from Parker, 1976 and Lai et al., 2017).**

Second, we demonstrate that active braiding intensity ($BI_A$) is proportional to both dimensionless sediment-stream power ($\omega^{**}$) and dimensionless stream power ($\omega^*$), despite the dimensionless critical time ($t_c^*$) of submarine and fluvial braided channels are different (Fig. 16). The dimensionless critical time is defined as $t_c^* = t_c Q_{in}/d_s^3$ (Lai et al., 2017). We found that the $t_c^*$ of lateral confined or unconfined submarine braided channels all fall around $10^8$ to $10^9$ (Table 2 and Fig. 16a); while the fluvial braided rivers fall around $10^{11}$. We think our smaller flume size, steeper bed slope and lighter bed material together make the

submarine braided channels to reach $t_c^*$ earlier than that of fluvial braided rivers. In addition, the Shields parameter ($\theta$) corresponding to our sediment is about 0.46, which is much larger than the critical threshold for transport (~0.04 to 0.06) and larger than occurs in the fluvial braided rivers (Egozi and Ashmore, 2009) (Fig. 15). The dimensionless sediment-stream power, defined as the combination of width-to-depth ratio ($B/h$), sediment-to-inflow discharge ratio ($Q_s/Q_{in}$) and bed slope ($S$) (Fig.

16b, modified from Paola, 2001), is proportional to confinement width, sediment supply and bed slope. When $Q_s$ is fixed, a

larger $Q_{in}$ reduces $\omega^{**}$, making the corresponding $BI_A$ become smaller. For instance, in Series B ($q_{in} \cong 84$ mm²/s, $Q_{in}/Q_s \cong$ 90), the submarine braided channels widen the imposed confinement width to accommodate the larger inflow, making the area of erosion and deposition in the DoDs more continuous and contiguous. This explains why increasing $Q_{in}$ slightly reduces the $BI_A$ of Series B at stable phase. The $\omega^{**}$ is applicable to both unconfined and confined submarine braided channels (Fig. 16b). On the other hand, the dimensionless stream power ($\omega^*$) considers density differences of the underflow, grain size ($d_s$) and

settling velocity ($\omega_s$) (Fig. 16c), but does not include the effects of $B$ and $Q_s$. The $\omega^*$ works well for both unconfined fluvial braided rivers (Zanoni et al., 2007; Egozi and Ashmore, 2009; Bertoldi et al., 2009) and confined fluvial braided rivers (Garcia Lugo et al., 2015)(Fig. 16c). In short, both $\omega^{**}$ and $\omega^*$ are proportional to $BI_A$. For an inverse problem, if $BI_A$ is known, these two relationships can be used to estimate the corresponding parameters (e.g., $h$, $Q_s$, $Q_{in}$, density differences), which provides a way to apply this work to paleoflow reconstruction.


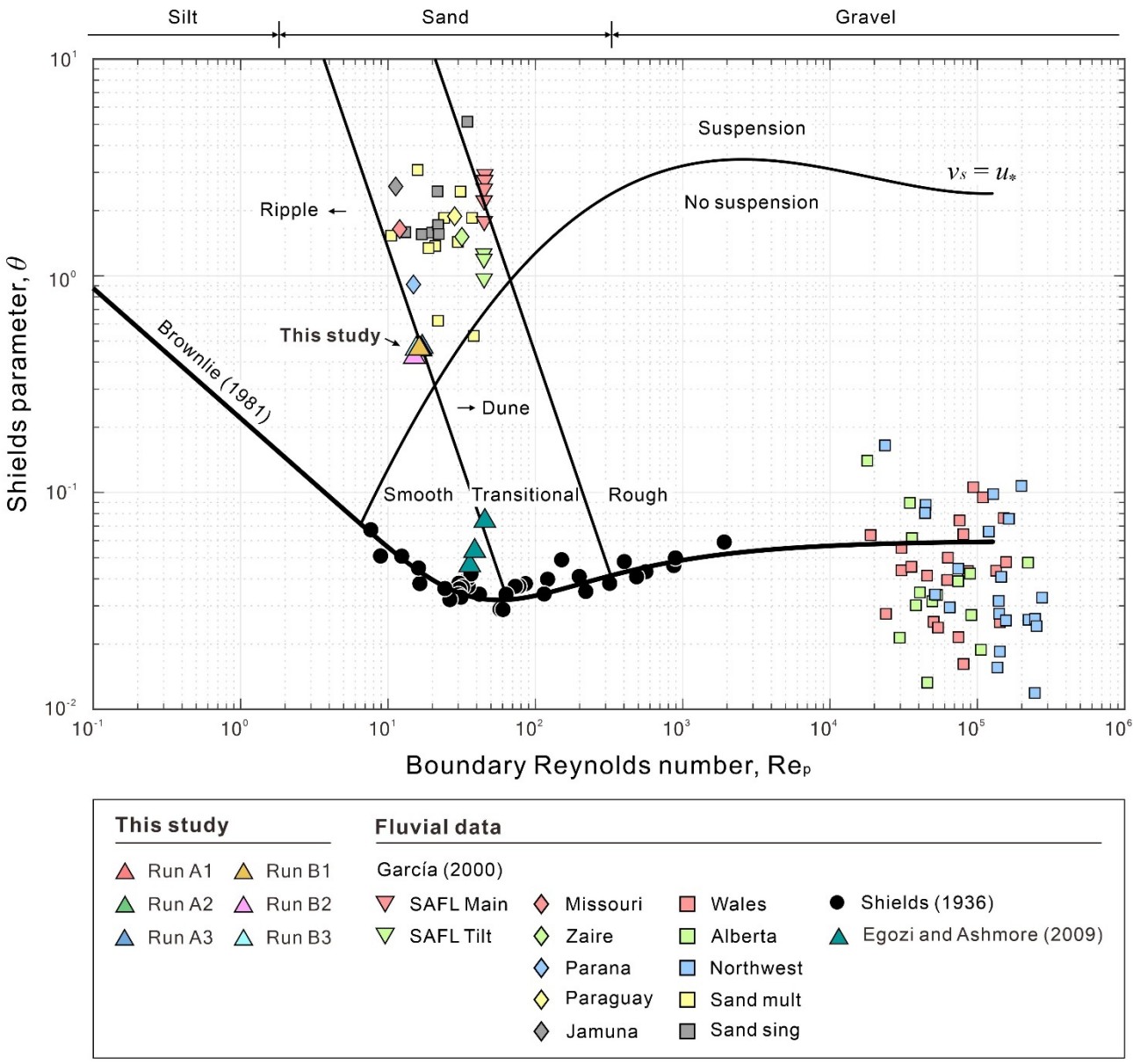

**Figure 15. Shields diagram of fluvial rivers and submarine channels (modified from García, 2000).**

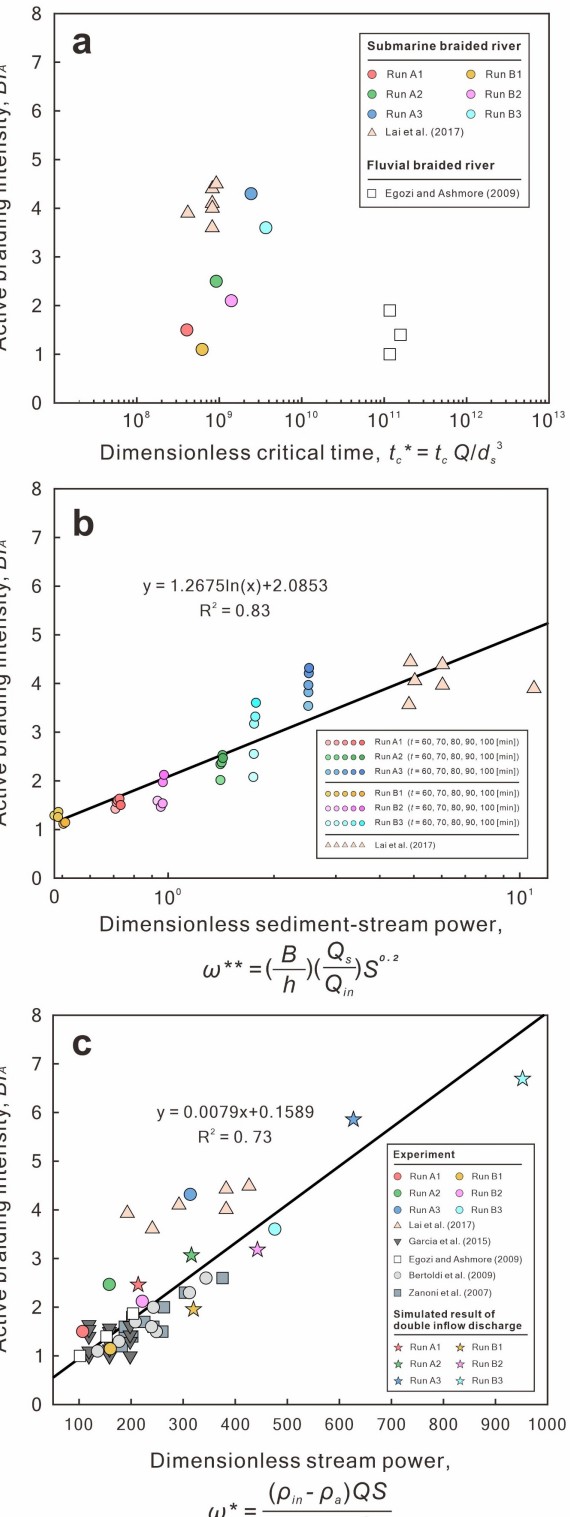

**Figure 16. (a)** Dimensionless critical time ($t_c^*$) of submarine and fluvial braided channels; **(b)** Relation of dimensionless sediment-stream power ($\omega^{**}$) to active braiding intensity ($BI_A$) for two sets of submarine braiding experiments; **(c)** Relation of dimensionless stream power ($\omega^*$) to active braiding intensity ($BI_A$) for two sets of submarine braiding experiments, four sets of fluvial braiding experiments and one set of our reduced-complexity model.

Until now, the dimensionless parameters ($t_c^*$, $\omega^{**}$, $\omega^*$) calculated from experiments of submarine and fluvial braided channels show strong linear relationships and support us to infer that the submarine braided channels on Baia di Levante Fan would continue to develop and prograde downstream. However, we think that the downstream lateral confinement may force the braided channels to transform into a single channel. The confinement width of these submarine braided channels gradually decreases from upstream to downstream and is well bounded by two erosive banks. The Baia di Levante Fan was formed by gradual stacking of gravity flows fed by repeated small-scale mass-wasting events rather than a single catastrophic collapse (Romagnoli et al., 2012). This modern field example agrees with the formation and morphological features observed in our laboratory experiments. Additionally, recent studies on the submarine flanks of La Reunion also confirm that long-term erosive processes combined with high sediment supply can form wide volcaniclastic deep-sea fans similar to the siliciclastic ones (Saint-Ange et al., 2011; Sisavath et al., 2011; Casalbore et al., 2021). Based on the reported bathymetry (Fig.1b), when the water depth is 1000 m, the bed slope of Baia di Levante Fan is 4 degrees, the confinement width is 3500 m (Romagnoli et al., 2012), and the number of submarine braided channels is about 3. Assuming these three submarine channels are active, we obtain $BI_A = 3$. Through the experimental scaling relationship (Fig. 16b), we obtain the corresponding dimensionless sediment-stream power is $\omega^{**} = 2.5$. Therefore, we can estimate the reasonable flow depth range of the turbidity current by assuming $Q_{in}/Q_s$. For instance, when $Q_{in}/Q_s = 13$, the estimated flow depth is 63.2 m. This depth agrees with the reported depth of the turbidity current (60 m) and the height of erosive walls (70−80 m) (Romagnoli et al., 2012). When $Q_{in}/Q_s = 9$, the estimated flow depth is 91.4 m. It is possible to cause overspill and form sediment waves (or cyclic steps) outside the bank. Similar morphology appears on the left outer bank of Baia di Levante submarine fan. Overall, the morphologic estimates derived from our proposed scaling relations are consistent with field observations, using plausible values for the inflow-to-sediment discharge ratio. Constraining these inputs for field cases would be major step toward predicting the morphology of submarine braided channels.

Additionally, the experimental results are intriguing in that they further emphasize the similarities in the propensity for and properties of braiding in fluvial and submarine systems (Lai et al., 2017). In isolation, the experimental results would predict braided submarine channels are common in modern oceans, however, they are exceedingly rare (Foreman et al., 2015). Herein, we built upon previous experiments that documented the same relationship between dimensionless stream power and active braiding intensity among rivers and submarine channels (Lai et al., 2017) and expanded the experiments to show both the active width and bulk morphologic change increase as the confinement width ($B$) increases commensurate with experimental fluvial systems. Thus, braided submarine channels can be spontaneously generated both through self-channelization on an

initially flat surface (Foreman et al., 2015; Lai et al., 2017; Limaye et al., 2018) and within a pre-existing confined channel (this study). Yet in nature both these bathymetric cases overwhelming result in sinuous, single-thread submarine channels. This implies there may be a missing attribute within the experiments that in nature suppresses braiding. Lai et al. (2017) suggested the missing attribute was enhanced deposition of suspended sediment loads within inactive channel threads during low-flow conditions. In nature higher sediment concentrations within turbidity currents as compared to river fluid flows likely

facilitate greater aggradation and in-filling of inactive channel threads. This process would limit segmentation of the flow by bars and promote development of a single thread over multiple threads. In the experiments there is no suspended sediment load. This phenomenon may be exacerbated in field submarine channels whose active width is confined by canyon or valley walls as compared to unconfined systems. In confined systems the fine-grained turbidity currents that overspill channel margins cannot be advected across the overbank area to zones more distal from the channel. Instead, the fine-grained material

is likely retained and forced to deposit within the active width. Interestingly, the Baia di Levante Fan is comparatively coarse in grain size distribution (Ramagnoli et al., 2012), and in the absence of significant silt and clay components may represent a unique case in which a braided system may evolve. This suggests that provenance and grain size distribution of sediment supply may play a nontrivial role in the submarine channel planform morphology. Finally, the observation that as the confinement width, $B$, increases the bulk morphologic change increases provides a testable hypothesis for stratigraphic studies

of submarine channels. Namely, increased bulk morphologic change should produce more heterogeneous channel sand-body deposits, particularly related to the abundance bar reactivation surfaces. It may be productive for future stratigraphic studies to quantify this abundance scaled to the channel geometry.

## 5 Conclusions

Physical experiments and a reduced-complexity model allow us to better explain the responses of submarine braided channels

to confinement widths and inflow-to-sediment discharge ratios. The rules of stream power are applicable to field-scale submarine braided channels. Our main findings are:

1.  We confirm that a larger confinement width ($B$) postpones the critical time ($t_c$) and increases the active braiding intensity ($BI_A$) at the stable phase. At a fixed confinement width, a larger inflow-to-sediment discharge ratio ($Q_{in}/Q_s$) slightly decreases the $BI_A$.

2.  The active width ($W_a$) and bulk change ($V_{bulk}$) are proportional to confinement widths for submarine braided channels, in agreement with fluvial braided rivers.

3.  The experimental and modeling results agree that active braiding intensity is proportional to both dimensionless sediment-stream power ($\omega^{**}$) and dimensionless stream power ($\omega^*$).

4.  Kernel density estimation (KDE) helps to visualize and predict the pattern of cross-sectional discharge distribution.

5.   We infer that the submarine braided channels on Baia di Levante Fan may continue to develop and prograde downstream, but that the downstream lateral confinement will eventually force the submarine braided channels to transform into a single channel.

     6.   Overall the relations we develop from the experimental and modeling results are able to predict the first-order general morphological and sedimentological patterns for field scale submarine fans.

## Data Availability

Data are available at: https://doi.org/10.5281/zenodo.7601496

## Supplement

The supplement related to this article is available online at:

## Author contributions

SYJL conceived the idea, SYJH performed the experiment and conducted the analysis. SYJL and SYJH drafted the paper, with contributions from ABL, BZF and CP. All authors worked on the final version submitted.

## Competing interests

The authors declare that they have no conflicts of interest.

## Acknowledgements

This study was supported by the Ministry of Science and Technology (MOST), Taiwan (grants to S. Y. J. L.: MOST 109-2628-E-006-006-MY3). Students of Morphohydraulics Imaging Laboratory (MIL) are well acknowledged during performing the experiments.

## Financial support

This study was supported by the Ministry of Science and Technology (MOST), Taiwan (grants to S. Y. J. L.: MOST 109-
2628-E-006-006-MY3).

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
