# Peer review of "Confinement width and inflow-to-sediment discharge ratio control the morphology and braiding intensity of submarine braided channels: Insights from physical experiments and reduced-complexity models"

_Earth Surface Dynamics, 2023_

## Author Comment (AC1)

**Responses to Review Comments**

In the following, review comments are in *blue italic* font, while responses are in **black normal** font.
* * *
**Reviewer #1 (David Nworie)**

*Dear Editor,*

*I have reviewed the paper by Huang et al titled "Confinement width controls the morphology and braiding intensity of submarine braided channels: Insights from physical experiments." It is well written and has great figures. I recommend publication after minor revisions. A few comments are below for the authors to consider addressing, in no particular order:*

**Reply:** We thank the reviewer's positive feedback and comments. Our responses to each question are listed below.

*(1) Title can be rephrased as "Confinement width controls the morphology and braiding intensity of submarine braided channels: Insights from physical experiments and reduced complexity models".*

**Reply:** We changed the title to "Confinement width and inflow-to-sediment discharge ratio control the morphology and braiding intensity of submarine braided channels: Insights from physical experiments and reduced-complexity models".

*(2) Line 86: Please include the concentration of sediments/particle (in percentage) in the dyed and saturated inflows for easy comparison with other experiments.*

**Reply:** In our experiments, we used "saturated brine" to simulate turbidity currents and we did not add fine grains for suspended load. The dry sediment added upstream is used to simulate "bed load." The saturated brine used in our experiments implies that the suspended load is always suspended in the flow layer and will bypass the basin without settling. These saline underflows can effectively transport bed load, change the morphology and form submarine braided channels. As requested, we can convert the inflow density ($\rho_{in}$= 1200 kg/m³) to sediment concentration by the following formula (Morris and Fan, 1997):

$$C = (\rho_{in} - \rho_w)/(1 - \rho_w/\rho_s) \qquad (1)$$

Therefore, the converted sediment concentration of the fluid is $C$ = 321.2 g/L. For reference, when the inflow density is $\rho_{in}$ = 1025 kg/m³, the corresponding sediment concentration of the fluid is $C$ = 40 g/L. Above this threshold, it is the typical condition for forming hyperpycnal flows in the ocean.

*(3) Describe the nature of the flows in the experiments in the result section. Where the flows turbulent or tractional or both? Relate this to your statement in line 365 "absence of suspended load" and 87.*

**Reply:** We use the essential dimensionless parameters to describe the characteristics of flow and sediment, including: Reynolds number ($Re$), densimetric Froude number ($Fr_d$), reduced bed shear stress ($\tau_b$), Shields parameter ($\theta$), boundary Reynolds number ($Re_p$), dimensionless stream power ($\omega^*$) and dimensionless sediment-stream power ($\omega^{**}$) (see Table 2 and Line 321 to Line 324). The dimensionless numbers are defined as follows:

| | | |
|---|---|---|
| Reynolds number | $$Re = \frac{hu}{v} = \frac{Q}{v}$$ | (2) |

| | | |
|---|---|---|
| Densimetric Froude number | $$Fr_d = \frac{u}{\sqrt{g'h}}$$ | (3) |

| | | |
|---|---|---|
| Reduced bed shear stress | $$\tau_b = (\rho_{in} - \rho_a)ghS$$ | (4) |

| | | |
|---|---|---|
| Shields parameter | $$\theta = \frac{\tau_b}{(\rho_s - \rho_{in})gd_s} = \frac{(\rho_{in} - \rho_a)ghS}{(\rho_s - \rho_{in})gd_s} = \frac{hS}{R'd_s}$$ | (5) |

| | | |
|---|---|---|
| Boundary Reynolds number | $$Re_p = \frac{u_*d_s}{v}$$ | (6) |

| | | |
|---|---|---|
| Dimensionless stream power | $$\omega^* = \frac{(\rho_{in} - \rho_a)QS}{\rho_{in}w_sd_s^2}$$ | (7) |

| | | |
|---|---|---|
| Dimensionless sediment-stream power | $$\omega^{**} = \left(\frac{B}{h}\right)\left(\frac{Q_s}{Q_{in}}\right)S^{0.2}$$ | (8) |

where $h$ is the flow depth of the saline underflow, estimated from the experiments ($h \approx 0.002$ m); $u = Q_{in}/(hB)$ is the cross-sectional averaged flow velocity; $B$ is confinement width; $v$ is water kinematic viscosity ($v = 10^{-6}$ m$^2$s$^{-1}$ of water $20°$ C); $g' = g(\rho_{in} - \rho_a)/\rho_{in}$ is the reduced gravity; $\rho_{in}$ is the density of inflow ($\rho_{in} = 1200$ kg/m$^3$); $\rho_a$ is the density of ambient water ($\rho_a = 1000$ kg/m$^3$); $S$ is bed slope; $\rho_s$ is the density of plastic sand ($\rho_s = 1500$ kg/m$^3$); $d_s$ is sediment grain size ($d_s = 0.34$ mm); $R' = (\rho_s - \rho_{in})/(\rho_{in} - \rho_a)$; $u_* = \sqrt{\tau_b/(\rho_{in} - \rho_a)}$ is shear velocity; $w_s = \frac{Rgd_s^2}{C_1v + (0.75C_2Rgd_s^3)^{0.5}}$ is sediment settling velocity reported by *Ferguson and Church* [2004]; $R = (\rho_s/\rho_{in} - 1)$ is submerged relative density of sediment ($R = 0.25$ in this study); $C_1 = 18$ and $C_2 = 1$ are two constants for typical natural sands.

Based on our calculations, the flow condition of the submarine braided channels is laminar (Re<200), subcritical flow (Fr<1), and the Shields parameter is around 0.4, which is much higher than the critical threshold (0.04~0.06). Conventionally, $Re_p$<5 implies the regime of smooth bed with small local flow velocity around the particle; $Re_p$>70 is the regime of rough bed with large local velocity. Our conditions ($Re_p \cong 16$) are in-between. We also add new Fig. 14 and Fig. 15 in the revised manuscript for better comparison between submarine braided channels and fluvial rivers. Above descriptions are all integrated in Table 2.

*(4) In the discussion, relate the influence of flow concentration to support mechanisms for transport and deposition and how does that relate to the braiding intensity.*
**Reply:** In this study, we use a fixed inflow density ($\rho_{in} = 1200$ kg/m$^3$), which corresponds to a fixed sediment concentration of the fluid ($C = 321.2$ g/L). Therefore, in this study we cannot demonstrate the influence of different inflow densities on submarine braided channels. However, according to Lai et al. (2017), active braiding intensity ($BI_A$) is proportional to the dimensionless stream power ($\omega^*$). Therefore, higher inflow density would result in higher active braiding intensity. Our current study indicates this trend still holds for both submarine braided channels and fluvial braided rivers with lateral confinements (see Fig. 16c).

*(5) The figures are clear and nice but some of the captions are short and say little about what is shown in the figures. Normally the Figures + Captions should be self-explanatory. I suggest the captions be expanded.*

**Reply:** We rewrote most of the figure captions to include more information.
* * *
**References cited in our reply:**

Foreman, B. Z., Lai, S. Y. J., Komatsu, Y., and Paola, C.: Braiding of submarine channels controlled by aspect ratio similar to rivers, Nat. Geosci., 8, 700-703, https://doi.org/10.1038/ngeo2505, 2015.

Lai, S. Y. J., Hung, S. S. C., Foreman, B. Z., Limaye, A. B., Grimaud, J. L., and Paola, C.: Stream power controls the braiding intensity of submarine channels similarly to rivers, Geophys. Res. Lett., 44, 5062-5070, https://doi.org/10.1002/2017GL072964, 2017.

García, M.: Discussion of "The Legend of AF Shields", J. Hydraul. Eng., 126, 718-720, 2000.

Morris, G. and Fan, J.: Reservoir sedimentation handbook, McGraw-Hill, New York, 1997.

Parker, G.: On the cause and characteristic scales of meandering and braiding in rivers, J. Fluid Mech., 76, 457-480, https://doi.org/10.1017/S0022112076000748, 1976.

---

## Author Comment (AC2)

**Responses to Review Comments**

In the following, review comments are in *blue italic* font, while responses are in **black normal** font.
* * *
**Reviewer #2 (Anonymous Referee)**

*(1) The primarily reports the results of physical experiments on the effects of channel confinement and sediment input ratio on the characteristics of braided channel patterns formed by submarine turbidity flows. In addition, results are supported by the output from a cellular model of turbidity flows based on caesar-lisflood to model the cross-channel distributions of depth, discharge and sediment flux. The results are compared with fluvial braided systems and shown to behave in similar ways with respect to confinement effects on braiding intensity, active width and volumetric erosion-deposition patterns. This has larger significance in relation to the physics of these flows and the potential to transfer understanding and relationships between these two environments and process-regimes. This also adds to results of Lai et al. 2017 and Limaye et al. 2018 and the paper demonstrates implications for full scale submarine braided turbidity flow channels and related sedimentology.*

**Reply:** We thank the reviewer's positive feedback and constructive comments. Our responses to each question are listed below.

*(2) Major suggestions:*

*(2.1) The role of B (confinement width) and Qin/Qs (sediment concentration or discharge/sediment load ratio) are emphasised throughout in relation to both braiding intensity and bulk volumetric change and active width. The experiments are designed to show especially the effect of confinement width on braiding intensity of turbidity flows. The design of the experiments also means that tests differ in total discharge and, consequently, discharge per unit width (Qin/B).*

**Reply:** Before answering the questions, allowing us to clarify the definition of inflow width ($b$), confinement width ($B$) and valley width ($W$) (see the updated Fig. 2). We also add these widths and the correct inflow unit width discharge ($q_{in}$) in the updated Table 1.

First, while we understand the logic of dividing the inflow total discharge ($Q_{in}$) by the confinement width ($B$) to estimate the inflow unit width discharge ($q_{in}$), we think that approach would be inconsistent with our observation that in the experiments, the submarine braided channels do not occupy the entire confinement width. Therefore, we instead calibrate the inflow unit width discharge by $q_{in} = Q_{in}/b$, where $b$ is inflow width (not confinement width $B$) (see the updated Fig. 2 and Table 1).

Second, we control the inflow-to-sediment discharge ratio ($Q_{in}/Q_s$) and inflow unit width discharge ($q_{in}$) at the same time. It may not be possible to produce submarine braided channels only by providing correct discharge per unit width without setting the correct inflow-to-sediment ratio. Based on previous reported successful runs of Foreman et al. (2015) and Lai et al. (2017), $Q_{in}/Q_s$ = 60 and 90 are the two reasonable values of this ratio.

Third, we set confinement width ($B$) as one of the main variables in this study for possible exploration in the future (which would not be limited to an initially straight channel): (1) If $B$ is gradually changed (widened, narrowed or irregular), how $B$ may influence the morphology of submarine braided channels? (2) When the left and right sides of the channel are confined bedrock, i.e., confinement width equals to valley width ($B = W$) or partly bedrock confined, how will this situation affect the development of submarine braided channels? These parts are beyond the scope of this study but we plan to address them in another paper.

*(2.2) Based on Table 1, Series A experiments have unit discharge of 40-50, while Series B experiments are in the range 60-85). If the width of the flows is also directly dependent on discharge and discharge per unit width, then there is, in addition to confinement width alone, a potential effect of discharge in the results. An example of this is the comparison of A2 and B1 which have the same discharge but A2 has two-times the confinement width of B1.*

**Reply:** The inflow unit width discharge ($q_{in}$) of Series A (Runs A1, A2 and A3) is controlled around $q_{in} \cong 55$ mm$^2$/s; while the inflow unit width discharge of Series B (Runs B1, B2 and B3) is controlled around $q_{in} \cong 84$ mm$^2$/s (about 1.5 times larger) (see Table 1). Although Run A2 and Run B1 have the same inflow total discharge ($Q_{in}$), the $q_{in}$ value for Run B1 is 1.5 times larger than for Run A2, while $b$ for Run B1 is 1.5 times smaller than for Run A2 (i.e., they do not differ by a factor of 2).

*(2.3) The interpretation ascribes the difference in BI to difference in confinement width but is it also potentially the result of the difference in discharge. The experiments assume fixed confinement width (although there is some erosion in some tests, as pointed out in the paper) and presumably differing total discharge and unit discharge are relevant to the outcome in channel network configuration as well as volumetric changes and active width.*

**Reply:** The experimental conditions listed in Table 1 should be read as: In Series A ($q_{in} = 55$ mm$^2$/s, $Q_{in}/Q_s = 60$), we want to compare the effect of confinement width ($B$) among Runs A1, A2 and A3; In Series B ($q_{in} = 84$ mm$^2$/s, $Q_{in}/Q_s = 90$), we want to compare the effect of confinement width ($B$) among Runs B1, B2 and B3. When comparing across Series A and Series B, we want to discuss the effect of $q_{in}$ or $Q_{in}/Q_s$ under the same confinement width ($B$).

*(2.4) A larger discharge in the same width, is, in effect, a relative confinement if the channel is not allowed to widen to accommodate the increased flow. If this idea is correct, some further analysis might be useful to look at this discharge effect, or to interpret the existing results in this way. It seems potentially the case that the differences between the A and B series are at least partially related to this effect but the text emphasises B and Qin/Qs as the explanation for differences.*

**Reply:** In our experiment, the channel is allowed to widen laterally (not limited by bedrock). The reason is the same as our reply in question (2.1). For future studies, we plan to explore the effects when the confinement width equals to valley width ($B = W$) or the channel configuration is partly bedrock confined with irregular shapes. In the current experiment, the two sides of confinement width are straight, erodible floodplains or terraces, not bedrock.

In Series A ($q_{in} \cong 55$ mm$^2$/s, $Q_{in}/Q_s \cong 60$), most of the submarine braided channels develop within the given confinement width without much widening to accommodate the inflow. However, in Series B ($q_{in} \cong 84$ mm$^2$/s, $Q_{in}/Q_s \cong 90$), the submarine braided channels would widen the given confinement width to accommodate the larger inflow and cause stronger bank erosion. We add this point to the Result section (see Line 239 to 242).

*(2.5) A similar argument might be made for the amplitude of topographic changes between runs (Fig 9) which are also ascribed to B but could also be a Qin effect. [As an aside, 'eyeballing' the DOD maps (Fig 8) seems to show that B2 has largest amplitude of change but this is not apparent in Fig 9].*

**Reply:** The DoD map (Fig. 8) presents the averaged sediment erosion and deposition pattern for $t =$ 4800 s to 6000 s of each experiment. Fig. 9 presents the homogeneous and heterogeneity (i.e., uniform or non-uniform) of DoD in terms of statistical distribution. We found that, under the same unit width discharge, the effect of $B$ on the heterogeneity of DoD is more significant, i.e., a smaller $B$ would result in more homogeneous DoD (e.g., Run A1 and B1); a larger $B$ would result in more heterogeneous DoD (e.g., Run A3 and Run B3). Although eyeballing the amplitude of topographic changes of Run B2 is the most significant, it is possible that a larger main channel appeared just in the last 1200 s, resulting in more drastic topographic changes. However, the statistical distribution shows that the DoD of Run B2 is relatively homogeneous.

*(3) The effect of confinement width on BIA is a major message of the paper but this relationship is not plotted in the paper. Figure 6 presents the time series of BIA for each experiment and this might be a good place to add a BIA vs B plot (and perhaps in relation to Q, Qin/B and Qin/Qs also?).*

**Reply:** Except for showing the relationship of $BI_A$ vs. $B$, $Q_{in}$, $q_{in}$, $Q_{in}/Q_s$, we list more essential dimensionless parameters in Table 2.

*(4.1) In places the discussion of the experimental results seems a bit cursory. I think it's a good idea to explain relevance/significance of some of these outcomes within the results. One example is the B-BIA results Lines 195-200, especially in the proposed effect of Qin/Qs in sand bar shapes etc. (which is presumably an important process-based explanation for the results also). Note also, the comment above on the Q-effect on these results.*

**Reply:** In Discussion section, we use dimensionless parameters to interpret the influence of physical parameters on the morphology of submarine braided channels. Although bar shape is a good way for process-based description, it is still controversial to generalize an objective indicator that can be applied to both submarine and fluvial braided channels. We use Table 2 to add more quantitative supports in our discussion section (see Line 321 to Line 324, Line 360 to Line 362 and new Fig. 14 and Fig. 15).

*(4.2) Another is around line 220 -225 in relation to erosion, widening and maps of erosion-deposition. These are intriguing results that are passed over quickly and also need a little more*

*justification, e.g., in the differences in channel erosion (widening) as a Qin/Qs effect rather than discharge alone being a cause of widening in some tests with larger discharge for the same confinement width in series A and B. For example, Qin/Qs cannot explain the clear differences in DOD patterns between B1 and B2 with almost identical Qin/Qs ratios. There is mention (line 225) of the areas of erosion-deposition becoming more "continuous and contiguous" but it would really help to support this claim (and its significance) in the analysis and text because it points to differences in processes and channel dynamics. This carries through to lines 235-240 where discussion and claims of influential variables are also a bit cursory.*

**Reply:** Runs B1 and B2 show the influence of channel primitive width ($B$) under the same $Q_{in}/Q_s$ and $q_{in}$ conditions. In this comparison, $BI_A$ is proportional to $B$ (see Table 2). The main reason for the bar shapes of Series B become more continuous and contiguous is due to the increased $q_{in}$ of Series B. As a result, many small channels merge into a few larger channels, making the $BI_A$ of Series B slightly decreased. This agrees with the dimensionless sediment-stream power ($\omega^{**}$). Under the same conditions of $B$, $h$, and $S$, the larger $Q_{in}/Q_s$ will make $\omega^{**}$ smaller, and the corresponding $BI_A$ will also become smaller. We update this point to the text (see Line 365 to Line 367).

*(5) The modeling results are very interesting and reveal patterns of flow and sediment transport that cannot be physically measured. They add insights that help to see the differences between test results and the possible flow-transport explanations for these differences. They also support some of the additional analysis in the discussion (e.g. Fig 14). It would help to connect the model results to the physical test results more explicitly and explain how this modeling supports and extends the physical tests.*

**Reply:** The direct comparisons between modeling results and physical experiments are shown in Fig. S11 to Fig. S16 (in the Supplement). Errors are controlled within an acceptable range (see Fig. S17). We decide to put these results in the Supplement in order to control the length of the manuscript. Therefore, we are confident to use the same calibrated parameters to predict the flow pattern under extreme events (i.e., double inflow total discharge). The results show that the linear relationship between dimensionless stream power ($\omega^*$) and $BI_A$ still holds for our model (see the star symbols in Fig. 16).

*(6) Fig 14b implies that B/h is a (the?) major control on BIA. Perhaps this suggests that B/h could be used earlier in the analysis of the experimental results? This would also be consistent with fluvial theory and observation.*

**Reply:** We agree that $B/h$ is indeed a simple and direct indicator, which can be used to classify first-order channel morphologies. We superimposed our experimental data in the modified Parker's diagram (see Fig. 14). The results agree with the theoretical prediction that smaller values of $h/B$ and larger values of $S/Fr_d$ result in more braids. Our results are consistent with those of Foreman et al. (2015) and Lai et al. (2017) for laterally unconfined submarine braided channels. These results are also true when comparing to field-scale fluvial rivers and submarine turbidity channels.

*(7) A question. Many fluvial braided relationships are from bedload dominated (often gravel) systems. Does the finer grained and suspension regime in turbidity flows make a significant difference to the processes and 'behaviour' of these braided systems?*

**Reply:** In our experiments, the plastic sand provided upstream of the water tank is used to simulate "bed load", so that our experimental submarine braided channels are the result of bed load. We use saturated brine to simulate turbidity currents, implying that the suspended load is completely dissolved in the saline underflow and will bypass the entire basin. Therefore, in our experiments, we cannot observe the contribution of fine sediment settled from the underflow. In terms of sediment transport, we add our experimental data in the Shields's diagram (see new Fig. 15). The result shows that sediment transport behavior of the submarine braided channels is similar to general criteria for fluvial rivers. In future work, it would be a potential topic to compare the difference between submarine braided channels formed by saline underflows and turbidity currents. If there is fine sediment that can settle, there should be a chance to observe the distribution pattern of fine grains.

*(8) And a few more minor suggestions:*
*Presentation of the experimental methods could include information on how sediment input was done and controlled along with experimental design on the basis for the choices of width, discharge and Qin/Qs ratio. The paper mentions D50 of the sediment but not whether it is uniform or not.*

**Reply:** The stable dry sand supply is controlled by a motor-driven conveyor belt. The values of $Q_{in}$ and $Q_{in}/Q_s$ are decided based on the past successful cases reported in Foreman et al. (2015) and Lai et al. (2017). When $q_{in}$ is determined, we can determine the amount of sand to be added by controlling the ratio of $Q_{in}/Q_s$. The choice of $B$ is based on our basin width ($W$ = 550 mm). In order to distinguish it from previous experiments (Foreman et al., 2015 and Lai et al., 2017) with full basin width, the way we choose $B$ is, $B$ = 0.12/0.55 = 22 % of $W$, 0.24/0.55 = 44 % of $W$ and 0.48 /0.55 = 87% of $W$, which represent the proportions of the confinement width occupying the valley width from small to large proportion, respectively. The sediment used in this study has a uniformity coefficient, $Cu$ = $d_{60}/d_{10}$ = 1.64 < 4 ($d_{60}$ = 0.46 mm, $d_{10}$ = 0.28 mm, $d_{50}$ = 0.34 mm), indicating that our plastic sediment is uniformly graded. We update the above descriptions to the Method section (see Line 86 to Line 89 and Line 95 to Line 99).

*(9.1) The automated channel mapping using dye intensity is very useful and gives a great visual impression of the channel pattern. From this the BIA is derived but the explanation of how active channel are defined (as opposed to identifying the dye channel threads that are not active) and extracted from these thresholds could be made more explicit. Or are all identified channels assumed to be active?*

**Reply:** We assume all captured channels are "active channels." The automated channels are verified by comparing them to those active channels observed in the experimental videos and time-lapse images. After the imaging thresholds are optimized, we then applied the same criteria to all images. According to our experimental observations, if a submarine channel is nearly semi-transparent (or presents in light green color), it probably has no ability to transport bed loads. Our current

automatic imaging method is not designed to distinguish those semi-transparent submarine channels.

*(9.2) Also, why is colour separation needed – in what way does that enhance the process and result?*

**Reply:** Using original color images, the quality and precision of the captured channel positions is unsatisfactory, i.e., there is too much noise. Our image processing workflow is to first convert a color image to an "enhanced gray-scale image" and then convert the enhanced gray-scale image into a binary image so that the channel positions and numbers can be captured automatically and precisely.

*(9.3) Using Fig 3 as an example, what automated values are used to identify channels and on what basis would that identify, for example, two channels rather than one in the centre of the middle cross-section? The automation is clearly a hugely useful technical step to increase sample size, but does it still require a decision about the threshold value?*

**Reply:** On the binarized image, the positions of channels should be white, i.e., the illumination value should be unity. When the channel has a discernable width, the corresponding brightness values would act like a top-hat function, which will make it difficult to capture the corrected channel positions. However, this kind of top-hat function can easily be diffused by a Gaussian filter and make the peak brightness easy for identification. Then the channel positions can be captured through a find peak function in Matlab. We use the built-in "findpeaks" function in Matlab. The more sensitive the parameters are set; the more channel numbers and locations will be captured. The parameter settings need to be calibrated with the experimental images for optimization. After that, this criterion is ready to be applied to all experimental images.

*(10) Qin/Qs is one of the main variables experimentally varied. It would help to explain a bit more in the introduction why this might be an important control and (in relation to 1) to say more about the choice to constrain it to essentially two values – one for series A (approx 60) and one for series B (approx. 90).*

**Reply:** Please see our reply to questions (2) and (8) for the setting of experimental conditions. These descriptions are integrated in the Method section.

*(11) "higher confinement width" is a bit distracting as a term – it means wider channel, but "higher confinement" might also imply narrower channel. Would using "higher width" instead be a problem?*

**Reply:** In order to better clarify the definition of various widths and avoid misunderstandings. We add inflow width ($b$), confinement width ($B$), valley width ($W$) on Fig. 2 and Table 1.

*(12) Some figure captions and labels could be changed to it easier to follow the results and data. Examples include Fig 3 where the images a-f could be labelled and/or explained in the caption. On Figs 6 and 8 it might help to label each with B, Q and Qin/Qs to avoid having to refer back to Table*

*1. On Fig 8 are the difference DEMs between time 0 and 6000? – it's not clear what times were used to do the differencing.*

**Reply:** We add the explanation to the figure caption of Fig. 3a to Fig. 3f. We label the experimental conditions and calculated time interval ($dt$ = 4800~6000 s) in Fig. 6 and Fig. 8. We rewrite most of the figure captions to include more information.

*(13) A small point on Fig 8; areas of bank erosion are picked out on some panels and mentioned in the text but erosion is also apparent in other cases e.g. B3?*

**Reply:** We mark the position where bank erosion occurs for each run in Fig. 8.
* * *
**References cited in our reply:**

Foreman, B. Z., Lai, S. Y. J., Komatsu, Y., and Paola, C.: Braiding of submarine channels controlled by aspect ratio similar to rivers, Nat. Geosci., 8, 700-703, https://doi.org/10.1038/ngeo2505, 2015.

Lai, S. Y. J., Hung, S. S. C., Foreman, B. Z., Limaye, A. B., Grimaud, J. L., and Paola, C.: Stream power controls the braiding intensity of submarine channels similarly to rivers, Geophys. Res. Lett., 44, 5062-5070, https://doi.org/10.1002/2017GL072964, 2017.

García, M.: Discussion of "The Legend of AF Shields", J. Hydraul. Eng., 126, 718-720, 2000.

Morris, G. and Fan, J.: Reservoir sedimentation handbook, McGraw-Hill, New York, 1997.

Parker, G.: On the cause and characteristic scales of meandering and braiding in rivers, J. Fluid Mech., 76, 457-480, https://doi.org/10.1017/S0022112076000748, 1976.

---

## Author Response (AR2)

**Responses to Review Comments**

In the following, review comments are in *blue italic* font, while responses are in **black normal** font.
* * *
**Associate Editor (Rebecca Hodge)**

*Thanks for submitting the revised version of your paper and for your detailed response to the reviewers. I have looked through the revised paper and have identified some places where it would benefit from some additional clarification (identified below). I would like you to address these points before the paper can be accepted for publication.*

*Best wishes,*

*Rebecca*

**Reply:** We thank the associate editor's positive feedback and comments. Our responses to each question are listed below.

*(1) Section 2.1: You provide a good explanation in the response to reviewers of why you use inflow width not confinement width to calculate unit discharge. However, I can't see where this is mentioned in the main text of the paper. (Readers might miss it if it's just in the table and figure.) I think that it would be useful to clarify this in the methods too (at around line 96 in the tracked changes version of the paper).*

**Reply:** In Line 98 to Line 102, we added the description for the calculation of inflow unit width discharge ($q$) and the selection of $Q_{in}/Q_s$.

*(2) Table 1: Is this the final bed slope? The methods say that the initial bed slope was the same in all experiments.*

**Reply:** Yes. In Table 1, the data is final bed slope. In order to distinguish "initial bed slope" and "final bed slope" more clearly, we add a column of initial bed slope in Table 1 and move the final bed slope to the left of "critical time". Please see the modified Table 1.

*(3) Section 2.3: You explain how the flow routing works, but I can't see where you explain how the model runs were carried out - e.g. what was the initial bed, what discharge was used, how many runs you did etc. Also, you should clarify that you're just modelling flow, not sediment erosion (especially as CAESAR can also model morphological change). You also need to explain that you performed some runs where you double the discharge; at the moment this just turns up in the results with no prior explanation.*

**Reply:** In Line 192 to 198, we add a description for the conditions of our simulations. In our model, we use the experimental DEM as the initial bed condition. The simulated inflow discharge is uniformly distributed at the upstream cells based on our experimental total inflow discharge. We mention that our model only simulates the underflows, i.e., the morphological changes are excluded. Additionally, we explain the condition of double inflow discharge which is used for testing whether the linear relationship still holds between dimensionless stream power and active braiding intensity under extreme events.

*(4) Are you actually using CAESAR-LISFLOOD if you aren't using the LISFLOOD flow-routing component of it?*

**Reply:** No, we did not use CAESAR-LISFLOOD in this article. In Line 152, we state that we developed a Matlab based reduced-complexity model, with modified hydraulic conditions for density currents. The numerical scheme is based on Thomas and Nicholas (2002) and the algorithm is according to the equation (2) to equation (8) in Section 2.3.

*(5) Section 3.3 and Fig 8: Why are the DoDs just shown for the final 20 min? How much bank erosion might have taken place in the previous timesteps?*

**Reply:** In our experiments, bank erosions formed gradually as the submarine braided channels developed. The eroded volumes depend on the chosen duration. In Fig. 8, we want to present the erosion-and-deposition maps and bank erosions at the same duration for submarine braided channels at the stable phase. The duration of the last 20 min achieves this purpose, which is better than showing the DoDs every 10 min or DoDs from $t = 0$ to 6000 s.

*(6) Fig 11: Does this show topography from the final timestep?*

**Reply:** Yes. In Fig. 11, we use the experimental DEM at the final stage for the initial bed surface in our simulations. We add a description in the figure caption.

---

## Author Response (AR3)

**Responses to Review Comments**

In the following, review comments are in ***blue italic*** font, while responses are in **black normal** font.
* * *
**Associate Editor (Rebecca Hodge)**

*Many thanks for making those edits to the paper. I'm happy to accept it for publication subject to you addressing one minor query as follows:*

*In Table 1, the initial and final slopes are given in different units. I'd suggest reporting them in the same units.*

*Thanks for submitting your work to Earth Surface Dynamics.*

**Reply:** We thank the associate editor's positive feedback and comments. We correct the unit of initial slope.
* * *
**Editor (Niels Hovius)**

*Dear Sam Y.J. Huang and colleagues,*

*Thank you for working with AE Rebecca Hodge and her referees during the review and revision of your manuscript. I am happy to follow her advice and approve publication of your work in ESurf. Please do address the issue in Table 1. I have one other request: the title may benefit from the redaction of the word 'braided'. Braiding is mentioned in direct connection with submarine channels. It is, in my opinion, not necessary to describe them as braided.*

*Your materials will now be prepared for 'print' by the publishers of Copernicus. Your prompt reaction to their queries and instructions will help keep the time to publication down.*

*Thank you for submitting this interesting study to ESurf. I trust that it will receive good attention.*

*Niels Hovius*

**Reply:** We thank the editor's positive feedback and comments. We correct the title to "Confinement width and inflow-to-sediment discharge ratio control the morphology and braiding intensity of submarine channels: Insights from physical experiments and reduced-complexity models".